# VAMP7-dependent late endosomal secretion of ER and mitochondrial proteins impacts the tumor microenvironment and macrophage engagement

Somya Vats[1], Pedro Dionisio[2,14], Quentin Lemercier[1,14], Raphael Pineau[3], Ludivine Therreau[4], Joanna Lipecka[5], Béatrice Cholley[1], Jean-Baptiste Moog[1], Jose Wojnacki[1], Céline Keime[6], Diana Zala[7], Philippe Bun[8], Sofia Freire[2], Neuza Domingues[2], Lydia Danglot[1,8], Ida Chiara Guerrera[5], Cédric Delevoye[9,10,11], Eric Chevet[3], Nuno Raimundo[2,12] ✉ & Thierry Galli[1,13] ✉

Late endosomal secretion is an unconventional secretion mechanism that depends on the SNARE protein VAMP7. We previously showed that VAMP7 mediates the secretion of the ER protein Reticulon3. However, the functional relevance and molecular mechanism of this secretory pathway remain unclear. Here, we show that VAMP7 knockout cells exhibit impaired secretion of ER- and mitochondrial-derived proteins and signs of ER and mitochondrial stress. In addition, pharmacological induction of organellar stress enhances the VAMP7-dependent secretion. We assess the pathophysiological significance of this mechanism using a preclinical glioblastoma model. VAMP7 knockout glioblastoma cells implanted in male rat brain develop into more necrotic tumors with reduced macrophage infiltration compared to controls, suggesting that VAMP7-dependent late endosomal secretion contributes to the tumor microenvironment and affects macrophage infiltration. Together, our results support a model in which late endosomal secretion functions as an organelle quality-control and stress-communication mechanism, with particular relevance to cancer.

Two main classes of protein secretion have been identified in eukaryotic cells. The most characterized one relies on protein insertion into the endoplasmic reticulum, transport through the Golgi apparatus, and secretion via vesicles generated in the trans-Golgi network. Alternatively, unconventional protein secretion (UPS) involves several mechanisms, with secretory vesicles originating from autophagosomes or endolysosomes. Secreted soluble proteins account for about 9–15% of the total human proteome and serve significant roles in cellular physiology, pathology, and intercellular communication. Most of the proteins secreted by UPS are leaderless proteins, i.e., they lack targeting signal sequences; however, they may include transmembrane proteins with or without signal sequences that pass the ER and reach the plasma membrane for secretion without going through the Golgi apparatus[1–4].

UPS can be triggered by cellular stressors, including nutrient, ER, mechanical, or inflammatory stress. We previously showed that the endosomal R-SNARE VAMP7 mediates the secretion of late endosomes, also called multivesicular bodies (MVBs), a major UPS pathway.

VAMP7 colocalizes with CD63, a marker of late endosomes and lysosomes, and mediates the secretion of CD63[5–9]. Structurally, along with the classical SNARE sequence with a SNARE motif and a C-terminal transmembrane domain, VAMP7 also possesses an N-terminal extension called the Longin domain[10], which plays an auto-inhibitory role by interacting intramolecularly with the SNARE motif[11]. This interaction inhibits the fusogenic activity of VAMP7[12,13]. VAMP7 mediates membrane fusion by interacting with target-SNAREs (t-SNAREs) located at the plasma membrane: Syntaxin1, Syntaxin3, SNAP23, and SNAP25, at the autophagosome: Syntaxin17 and SNAP29 and at the ER/ERGIC Syntaxin5 and SNAP47[14–16]. VAMP7 is involved in mediating endolysosomal secretion in fibroblasts[12,17–19] and in the lysosomal secretion of ATP in astrocytes[20] and epithelial cells[21]. VAMP7 is a key regulator of neuronal morphogenesis, as the VAMP7-mediated vesicular transport aids in the neurite outgrowth, which the overexpression of the auto-inhibitory Longin domain[12,22] can impede. The fusogenic activity of VAMP7 has also been implicated in degradative autophagy. It is involved in the fusion of ATG9-containing vesicles originating from the recycling endosomes, which help in the phagophore expansion[23,24] and in the autophagosome-lysosome fusion, particularly in *Drosophila*, where no homolog of the v-SNARE regulating autophagosome-lysosome fusion VAMP8 exists[25]. Importantly, VAMP7 and the t-SNAREs Syntaxin17 and SNAP29 mediate the fusion of mitochondrial-derived vesicles (MDVs) with endosomes and lysosomes[26], a process that is thought to mitigate small-scale mitochondrial damage. VAMP7 knockout (VAMP7KO) mice also showed mitochondrial impairment in insulin-secreting pancreatic β cells[24]. Accordingly, the Syntaxin17-SNAP29-VAMP7 complex was found to regulate mitophagy under severe hypoxia, thereby ensuring mitochondrial quality[26]. Our most recent work showed that late endosomal secretion is enhanced in the absence of degradative autophagy, i.e., in autophagy-null cells such as ATG5 knockout (ATG5KO) cells[27] in good agreement with observations made in neutrophils[28]. We analyzed the secretome of NGF-differentiated PC12 cells that had acquired a neuron-like phenotype. We found that the secretion of tubular ER-phagy-related LC3-interacting-region-containing protein Reticulon 3 (RTN3) was increased in ATG5KO and decreased in VAMP7KO cells. This led to the notion that autophagy-dependent late endosomal secretion, mediated by VAMP7, contributed to ER quality control during neurite growth[27]. Still, the whole mechanism could not be explored in neuronal cells.

Here, we generated fibroblastic and glioma cells devoid of VAMP7 to characterize the role of late endosomal secretion in non-neuronal cells. Both ER- and mitochondria-associated components were less secreted in VAMP7KO cells and more in ATG5KO cells, and this was affected by ER and mitochondrial stress, respectively. Given the unknown signaling nature of UPS, we reasoned that it might play a role in pathogenesis. Orthotopic syngeneic grafts of VAMP7KO rat glioma cells led to tumors with reduced macrophage infiltration compared to those formed by WT cells. We conclude that VAMP7-dependent late endosomal secretion operates the release of elements originating from the ER and mitochondria, depending on autophagy and stress (both often found activated in tumor cells). The released material thus provides signals associated with tumor growth attenuation, likely by reshaping the tumor microenvironment.

## Results

### VAMP7KO impairs the secretion of tubular ER and mitochondrial proteins by late endosomes

Following up on previous studies on autophagy-dependent late endosomal secretion in neuronal cells[9,27], we sought to characterize the roles of VAMP7 and autophagy in unconventional secretion in non-neuronal cells. To this end, we created Crispr-Cas mediated knockouts (KO) of VAMP7 or ATG5 in Normal Rat Kidney (NRK) cells (Fig. 1A). We found that secretion of tubular ER RTN3 and mitochondrial Voltage-Dependent Anion-selective Channel (VDAC, a marker of the

mitochondrial outer membrane) was strongly diminished in VAMP7KO and increased in ATG5KO NRK cells independently of Bafilomycin A1 (BafA1) treatment (Fig. 1B–E, Supplementary Fig. 1A). This extends a principle initially established in neuronal cells to a cell type not specialized in protein secretion. To further assess the properties of the cellular components released via VAMP7-dependent mechanisms, we isolated extracellular vesicles from wild-type (WT), VAMP7KO, and ATG5KO cells. We analyzed them by immunoblotting for the presence of the standard exosomal marker, CD63. Moreover, we performed Dynamic Light Scattering-based Nanoparticle Tracking Analysis (NTA) to measure the size and concentration of the released small (sEVs, < 200 nm diameter) and large (LEVs, > 200 nm) EVs. VAMP7KO cells were deficient in the secretion of CD63 and sEVs without any significant effect on LEVs as compared to WT and ATG5KO cells (Fig. 1F, G, Supplementary Fig. 1B). This result unequivocally positions VAMP7 as a main endosomal R-SNARE mediating the secretion of sEVs/exosomes in good agreement with recent work[9,19].

We then assessed whether VAMP7KO could affect the subcellular localization of CD63 in NRK cells. VAMP7KO cells exhibited an accumulation in peripheral CD63+ vesicles. Re-expression of GFP-VAMP7 rescued the subcellular localization of CD63 to the perinuclear region as found in WT cells (Fig. 1H, I). The link between VAMP7 and CD63 was further established by the detection of CD63 in GFP-VAMP7 immuno-precipitates using proteomics (Supplementary Data 3). Perinuclear CD63+ vesicles are thought to correspond to degradative compartments, whereas peripheral ones are believed to correspond to secretory late endosomes[29]. Accumulation of peripheral CD63+ vesicles suggested an inhibition of their secretion in VAMP7KO cells.

We next analyzed the effects of VAMP7KO on ER structure. In contrast to other conditions, the subcellular localization of RTN3 in VAMP7KO cells was similar to that previously described for CD63. RTN3 staining−i.e., increased at the cell periphery and decreased in the perinuclear region in VAMP7KO cells (Supplementary Fig. 1D, E)− suggests that VAMP7KO affects the subcellular distribution of RTN3 and CD63 in a similar manner.

To analyze the effect of VAMP7 on endolysosomes, we assessed lysosomal function by measuring lysosomal pH, which serves as a proxy for their degradative capacity[30,31]. As expected, peripheral lysosomes closer to the plasma membrane showed higher pH, indicating reduced degradative potential, independent of cell genotype (Fig. 1J, K). Mounting evidence supports different roles for lysosomes depending on their intracellular position: perinuclear lysosomes are widely regarded as more degradative and more acidic, while peripheral lysosomes may be more closely linked to other functions, including secretion[29,32,33]. These concordant experimental and literature-based observations may therefore suggest that the redistribution of CD63+ vesicles from the perinuclear region to the periphery in VAMP7KO NRK cells is consistent with the observed inhibition of sEVs and CD63 secretion observed in these cells. Collectively, these experiments show that VAMP7 is required for the secretion of sEVs (containing CD63), and for the release of RTN3 and mitochondrial proteins, suggesting that VAMP7-dependent late endosomal secretion is indeed involved in the release of ER and mitochondrial material.

### The loss of VAMP7 does not suppress degradative autophagy in NRK cells

The role of VAMP7 in autophagosome formation and maturation[23–25], and the role of autophagy in degrading ER and mitochondria have been previously described. To assess the effect of VAMP7KO on autophagy, we measured basal and induced autophagy in NRK cells. VAMP7KO cells showed an increase of LC3-I to LC3-II conversion; however, this was not due to a block of autophagosome-lysosome fusion because the levels of p62 did not change significantly in these cells (Supplementary Fig. 2A–D). We observed an initial change (0 h) in the LC3-I to LC3-II conversion with BafA1 treatment in these cells;

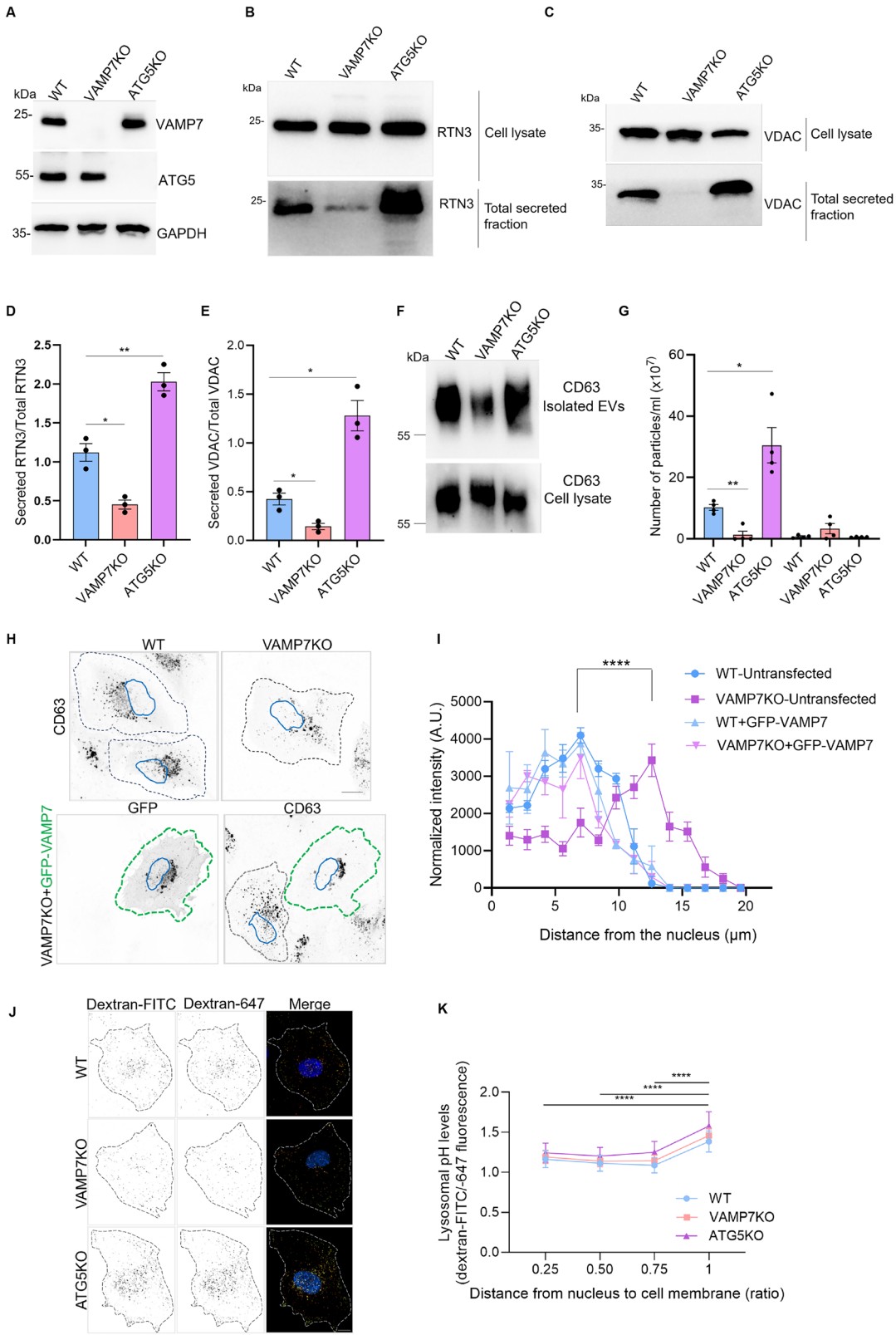

however, a time-course treatment did not show any significant alteration in autophagy between WT and VAMP7KO cells (Supplementary Fig. 2E, F).

To better analyze autophagic flux, we transfected cells with the tandem RFP-GFP-LC3 plasmid and either treated them with Torin1 to induce autophagy or left them untreated. We observed a slight change in the number of autophagosomes in Torin-treated WT vs. VAMP7KO

NRK cells; however, the number of autolysosomes remained unchanged. Torin1 treatment induced autophagy in a similar manner in both WT and VAMP7KO NRK cells, as visualized by the increase in both the number of autophagosomes and autolysosomes (Supplementary Fig. 2G–I). Hence, autophagy in NRK cells appeared slightly affected, rather than blocked, in autophagosome formation or fusion with lysosomes due to the lack of VAMP7, suggesting a limited role for late

**Fig. 1 | VAMP7KO impairs the secretion of tubular ER and mitochondrial proteins. A** WT, VAMP7KO and ATG5KO NRK cell lysates were blotted for VAMP7, ATG5 and GAPDH to confirm the KOs of VAMP7 and ATG5 in these cells. **B–E** WT, VAMP7KO and ATG5KO NRK cells were simultaneously treated with BafA1 (100 nM) to inhibit degradative autophagy and serum starved overnight. The next day, the secreted media was processed, and total secreted protein was precipitated using acetone. The total secreted fraction and cell lysates were collected and analyzed and the ratio of secreted RTN3A over total RTN3A or secreted VDAC over total VDAC was plotted. Two-tailed Welch's t-test was used to determine the statistical significance between three independent experiments. *$p$-val < 0.05, **$p$-val < 0.01. Data is the average of three independent experiments presented as the mean ± SEM. **F**, **G** WT, VAMP7KO and ATG5KO NRK cells were treated with BafA1 (100 nM) and serum starved overnight. The next day, the secreted media was processed using a commercial kit and exosomes were isolated. The properties and numbers of the secreted extracellular vesicles from WT, VAMP7KO and ATG5KO cells were analyzed using western blotting of the exosomal marker CD63 (**F**) and by using DLS Zetasizer (**G**) to measure the exosome size and number. Welch's t-test was used to determine the statistical significance. Data is derived from four independent experiments and is presented as the mean ± SEM *$p$-val <0.05, *$p$-val < 0.01. **H, I** WT

and VAMP7KO cells were either left untransfected or were transfected with GFP-VAMP7 and immunostained with CD63. A home generated ImageJ plugin was used to measure the normalized intensity of the CD63 signal. Statistics was performed using two-way repeated-measures ANOVA for three independent experiments for a total of 30 cells. Statistically significant ($p$-val < 0.0001) interaction and row factor values between untransfected WT and VAMP7 signifies a change in the distribution pattern of the CD63 signal with respect to the distance from the nucleus. Error bars represent the standard deviation (SD). **J** Peripheral lysosomes are less acidic, and therefore less degradative, in NRK cells of all genotypes. Representative images of live NRK WT, VAMP7KO and ATG5KO cells stained with Hoechst and previously incubated with dextran-FITC (pH sensitive) and dextran-647 (pH insensitive) for 16 h, followed by a washout of 6 h, to ensure that endocytosed dextrans were fully delivered to lysosomes. **K** Quantification of the mean dextran-FITC/dextran-647 ratio (higher level indicate higher pH and therefore reduced degradative capacity) for lysosomes, categorized by their relative proximity to the nucleus into four quartiles. One-way ANOVA with Bonferroni's post hoc test was used to determine the statistical significance between three independent experiments. ****$p$-val < 0.0001.

endosomes in these processes. We also did not observe any alteration in the mTOR signaling pathway, the upstream regulator of autophagy (Supplementary Fig. 2J–M). These results showed that VAMP7KO does not significantly alter autophagy, thereby ruling out a role for degradative autophagy in the tubular ER and mitochondrial secretion defects observed in VAMP7KO cells.

## VAMP7KO alters mitochondria and ER-associated gene expression

To evaluate the global cellular changes induced by VAMP7KO and ATG5KO, we compared the transcriptomes of those cells with that of WT NRK cells (Fig. 2A). We identified differentially expressed genes (DEGs) between the VAMP7KO and WT sample group and the ATG5KO and WT sample group. DEGs (Supplementary Data 1 and Supplementary Data 2, Supplementary Fig. 3A) were then analyzed for functional enrichment using Gene Set Enrichment Analysis (GSEA)[34]. The GSEA gene ontology dataset for cellular components showed enrichment of several mitochondrial components in the VAMP7KO, including the outer and inner mitochondrial membranes, mitochondrial matrix, and the ATP synthase complex. In contrast, functional enrichment in ATG5KO cells was more related to the ER lumen (Fig. 2B, Supplementary Fig. 3E). When the Hallmark Gene Set was used as a reference, the VAMP7KO dataset showed a positive enrichment for oxidative phosphorylation, the unfolded protein response (UPR), mTORC1 signaling, and cholesterol homeostasis. In contrast, the ATG5KO dataset showed enrichment for pathways related to inflammation, including the interferon alpha and gamma response pathway (Supplementary Fig. 3B). We also compared the VAMP7KO and the ATG5KO datasets using the same approach. We observed a positive enrichment of the UPR term in the VAMP7KO compared to ATG5KO (Supplementary Fig. 3C). The UPR is a stress-signaling pathway induced by cells to cope with modifications in ER homeostasis[35]. A positive enrichment of the UPR hallmark with an enrichment score of 1.82 was observed in VAMP7KO vs ATG5KO (Supplementary Fig. 3C, D). Interestingly, when analyzing the UPR-annotated genes, 11 of them were identified as potential ATF4 targets (as based on ChipAtlas data), and 5 of them as ATF6 targets (ChipAtlas and literature); only 3 were associated with XBP1s. Accordingly, we observed an increased expression of BiP (an ATF6 target[36] and an increased phosphorylation of eIF2α (the main target of the Integrated Stress Response (ISR)[37] in VAMP7KO cells compared to both WT and ATG5KO cells (Fig. 2C–E). Furthermore, confocal live-cell imaging of ER architecture using ER-Tracker revealed an increased occurrence of ER whorls in VAMP7KO cells, structures typically associated with ER stress[38] (Fig. 2F, G). Consistently, conventional transmission electron microscopy (TEM)

analysis confirmed the accumulation of ER whorls in VAMP7KO cells (Fig. 2H, I). Together with transcriptomic and protein expression data, these findings indicate that VAMP7 deficiency is associated with an alteration in global ER and mitochondria homeostasis.

Previous studies reported that mitochondrial function is impaired in VAMP7KO insulin-secreting cells and that autophagy regulates mitochondrial quality through mitophagy[39]. To further explore this, we analyzed WT and VAMP7KO NRK cells by TEM and observed a significant increase in both mitochondrial number and length in VAMP7KO cells compared to WT cells (Fig. 2J–L). Thus, we tested whether VAMP7KO and ATG5KO cells exhibit alterations in mitochondrial mass by staining cells with the mitochondrial reporter dye Mitotracker green, which accumulates in mitochondria independently of the inner membrane potential. We observed a robust increase in Mitotracker green intensity in both VAMP7KO and ATG5KO cells compared to the WT control (Supplementary Fig. 4A–C). We also observed an increase in mitochondrial mass by performing immunoblotting of the mitochondrial outer membrane protein VDAC (Supplementary Fig. 4D, E), one of the most abundant mitochondrial proteins. This is consistent with previously published data showing an increase in mitochondrial mass in ATG5KO T lymphocytes[40]. Next, to determine whether the increase in mitochondrial mass translates into changes in mitochondrial respiration, we measured oxygen consumption rate (OCR) using a widely accepted real-time respirometry approach[41]. Both VAMP7KO and ATG5KO cells showed a significant decrease in basal respiration, maximal respiration, and spare respiratory capacity as compared to the WT cells (Supplementary Fig. 4F). In addition, VAMP7KO cells appeared less affected than ATG5KO cells, which are deprived of mitophagy. These results show that, despite increased mitochondrial mass, mitochondrial respiration is decreased in VAMP7KO and ATG5KO cells. This suggests that the loss of either VAMP7KO or ATG5KO results in a defect in mitochondrial aerobic metabolism, possibly related to increased mitochondrial stress, consistent with the activation of the ISR and the phosphorylation of eIF2α[37].

Next, we performed a detailed analysis of mitochondrial structure in WT, VAMP7KO, and ATG5KO cells. We stained the cells with Mitotracker Deep Red and quantified the 3D-mitochondrial structure using the freely available Mitochondria Analyzer plugin in ImageJ (Supplementary Fig. 5A)[42]. VAMP7KO and ATG5KO cells showed changes in both morphological parameters (i.e., mitochondrial volume, surface area, and sphericity) and connectivity parameters (i.e., number of branches, total branch length, and junctions). Both VAMP7KO and the ATG5KO showed a significant increase in mean volume, mean surface area, and a decrease in sphericity, compared to the WT control

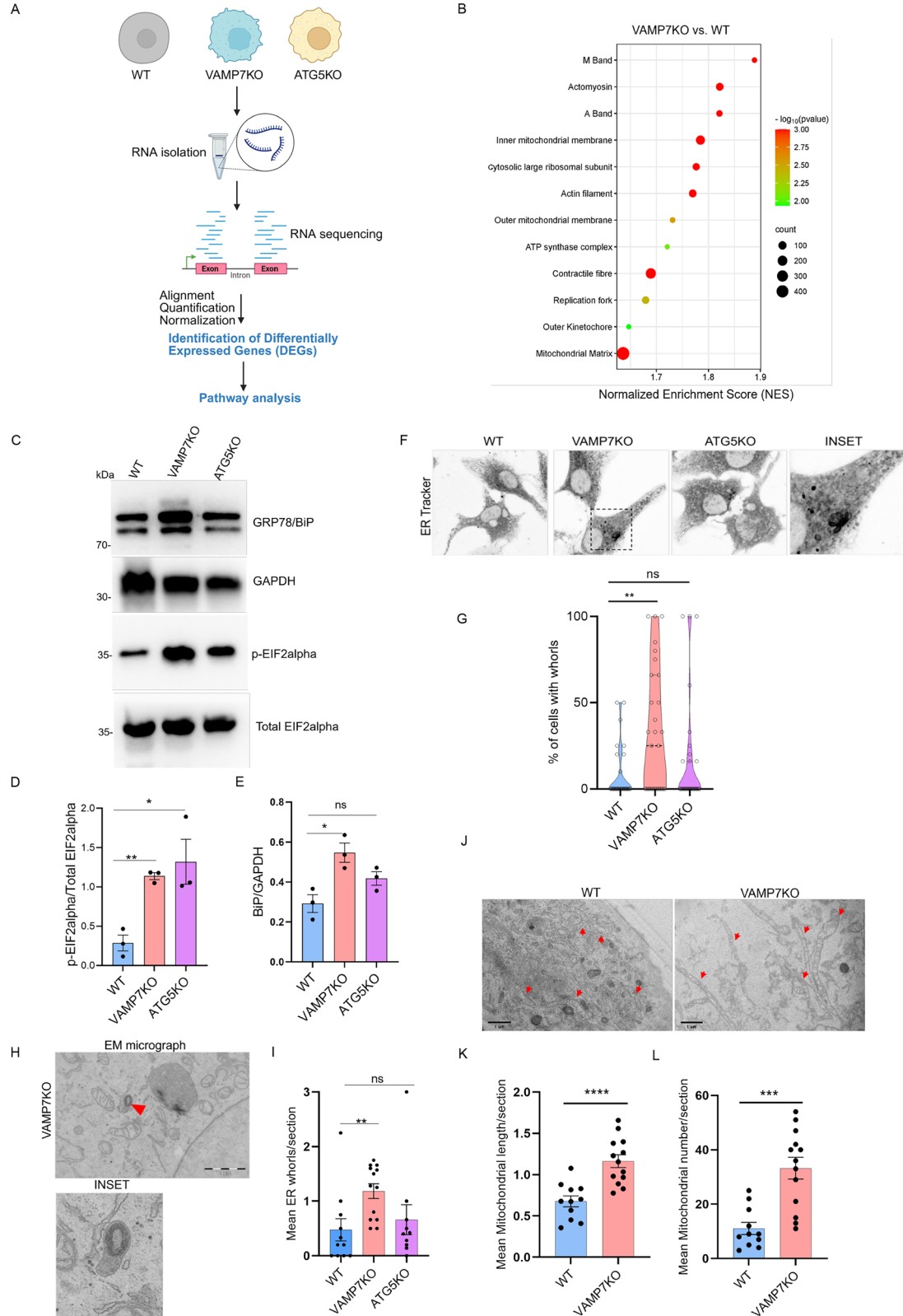

(Supplementary Fig. 5B–D). As per the connectivity parameters, the VAMP7KO and ATG5KO cells showed a significant increase in the number of branches, total branch length, and junctions compared to the WT control cells (Supplementary Fig. 5E–G). All quantitative analyses described above indicate a hyperfused mitochondrial network in both VAMP7KO and ATG5KO cells. Mitochondria constitutively undergo cycles of 'fission' and 'fusion', and MDV generation, in

processes tightly regulated by the physiological and pathological state of the cell. Mitochondrial hyperfusion has now been studied as an essential survival strategy for cells undergoing nutrient deprivation or stress such as ER stress. It can be due to increased fusion or decreased fission. Fission depends on recruiting Dynamin-like DRP1, the protein that catalyzes mitochondrial fission, to mitochondria by receptors such as the mitochondria-fission factor (MFF). Furthermore, DRP1 is

**Fig. 2 | VAMP7KO alters mitochondria and ER-associated gene expression.**
**A** Schematic representation of the RNAseq analysis workflow in which the differential gene expression between WT, VAMP7KO and ATG5KO NRK cells was examined, and pathway analysis was performed. The transcriptomics of each gene was analysed by four independent experiments. Scheme created in BioRender. Galli, T. (2026) https://BioRender.com/s8m3xp3. **B** Gene Set Enrichment Analysis (GSEA) performed for Gene Ontology (GO) pathways relative to Cellular Components (CC) shows a significant enrichment of mitochondria-associated proteins in the VAMP7KO dataset vs WT. p-values were determined according to analysis in the database for annotation, visualization, and enrichment analysis (Fisher exact *p*-value). **C–E** WT, VAMP7KO and ATG5KO NRK cell lysates were blotted for GRP78/BiP, p-EIF2α, EIF2α and GAPDH expression. The ratio of GRP78 over GAPDH or the ratio of p-EIF2α over total EIF2α was quantified and plotted. Two-tailed Unpaired t-test was used to determine the statistical significance for three independent experiments. *p*-val < 0.05, **p*-val < 0.01, ns: non-significant, error bars represent the standard error of mean (SEM). **F, G** Live microscopy of WT, VAMP7KO and ATG5KO NRK cells stained with ER Tracker. The graph depicts the percentage of

cells with ER whorls in each acquired image. Two-tailed Mann-Whitney test was used to determine the statistical significance. **p*-val < 0.01, ns non-significant, *n* = 30 images acquired per condition from three independent experiments. **H, I** WT and VAMP7KO NRK cells were imaged using a Transmission Electron Microscope. Red arrowhead indicates an ER whorl. Electron micrographs of ER whorls in WT and VAMP7KO NRK cells illustrate the upregulation of ER stress in VAMP7KO NRK cells. Data represents the mean number of ER whorls per section. Data is the average of three independent experiments presented as the mean ± SEM (WT, *n* = 11 cells; VAMP7KO, *n* = 13 cells, ATG5KO, *n* = 9 cells). Two-tailed Welch's t-test was used to determine the statistical significance between samples. **J–L** WT and VAMP7KO NRK cells were imaged using a Transmission Electron Microscope. Red arrows indicate mitochondria. Electron micrographs and quantification of mean mitochondrial length and number of mitochondria per section in WT and VAMP7KO NRK cells illustrate changes in mitochondrial length and numbers in VAMP7KO NRK cells. Two-tailed Welch's t-test was used to determine the statistical significance between samples. Data is the average of three independent experiments presented as the mean ± SEM (WT, *n* = 11 cells; VAMP7KO, *n* = 13 cells).

---

regulated by phosphorylation; some of these (e.g., Ser637) keep it away from mitochondria, while others (e.g., Ser616) target it towards the organelle. To probe the dynamics of the mitochondrial network in WT, VAMP7KO, and ATG5KO cells, cells were subjected to 2 h starvation in Earle's Balanced Salt Solution (EBSS), a stimulus known to trigger hyperfusion[38–41] followed by a 30-minute refeeding period. We observed a reduction in the levels of the activated (pro-fission) DRP1 phosphorylated at Ser616 and, to a lesser extent, of the total DRP1 in both VAMP7KO and ATG5KO cells (Supplementary Fig. 5H–J). Additionally, these cells showed reduced MFF levels (Supplementary Fig. 5K, L). These results suggest that multiple stimuli (less fission, increased mass) modulate mitochondrial structure, dynamics, and function in VAMP7KO and ATG5KO NRK cells but not in WT cells, possibly due to impaired quality-control mechanisms. Defects observed in VAMP7KO cells may correspond to a block of MDV biogenesis, while those in ATG5KO cells may most likely be linked to mitophagy defects. Given that ER and mitochondria are often functionally and physically associated and that coincident defects in tubular ER and mitochondria were observed in our study, we reasoned that VAMP7 might similarly regulate both intracellular compartments, converging towards the activation of common adaptive signaling pathways.

## VAMP7 mediates transport between the ER and CD63+ late endosomes

Our parallel observations of tubular ER and mitochondrial defects in VAMP7KO cells led us to investigate the molecular mechanism of tubular ER transport to late endosomes, based on the mechanism of fusing, by which MDV fuses with late endosomes in a VAMP7-dependent manner[26]. We designed two assays to measure the presence of mitochondrial and ER components in CD63+ late endosomes by colocalization and proximity to CD63. Firstly, we used confocal microscopy to visualize the association of ER and mitochondrial components with CD63, a marker of late endosomal/MVB compartments, as proxies for their transport to late endosomes. These compartments are the veritable carriers of unconventional secretory cargoes destined for the plasma membrane. We used RTN3 to stain the reticular ER and Mitotracker to visualize mitochondria. We observed increased co-occurrence of RTN3 and Mitotracker with CD63, as measured by Pearson's correlation coefficient, in ATG5KO cells, and this was further enhanced by BafA1 treatment in WT and ATG5KO cells Fig. 3A–D). To corroborate the increase in the association of ER and mitochondrial components with CD63+ compartments upon stimulation of VAMP7-dependent secretion, we performed proximity ligation assays (PLA) to quantify the proximity of endogenous proteins to VDAC and CD63, as well as RTN3 and CD63. WT, VAMP7KO, and

ATG5KO cells were either left untreated or treated with BafA1. The number of PLA-positive dots representing VDAC-CD63 and RTN3-CD63 proximity increased in ATG5KO cells and decreased in VAMP7KO cells (Fig. 3E–H). We further utilized RTN3 as a bona fide secreted protein via VAMP7-mediated mechanisms and employed super-resolution STED microscopy to image the vesicular compartments driving this secretion. Donut-shaped membrane compartments marked with CD63 were observed with close apposition of reticular RTN3 (Fig. 3I). Altogether, these data suggest a parallel, VAMP7-dependent mechanism that allows tubular ER and mitochondrial proteins to reach late endosomes.

## The short form of RTN3 is transported to CD63+ late endosomes and interacts with CD63 in a VAMP7-dependent manner

RTN3 is an ER protein that maintains the structural integrity of the tubular ER[43]. There are several RTN3 isoforms, amongst which the 135 kDa protein, herein identified as the long isoform, is the most studied. Work from the Dikic laboratory characterized the specific role of the long form of RTN3 in ER-phagy, the autophagic degradation of ER proteins[44]. The long isoform of RTN3 has several LC3-interacting regions, allowing it to bind to the autophagy protein LC3 and assist in packaging misfolded proteins from the ER into autophagosomes for degradative autophagy[44,45]. To characterize the RTN3 isoform being secreted through VAMP7-dependent secretion, we used the simultaneous expression of a Myc-tagged short form RTN3 (Myc-RTN3A) and a Flag-tagged long form RTN3 (Flag-RTN3L). We observed the specific secretion of the short form of RTN3 (RTN3A) but not of the long form (RTN3L) (Fig. 4A, B). We next investigated whether the secreted short form of RTN3 and CD63 were present in the same complex. We expressed Myc-RTN3A in WT, VAMP7KO, and ATG5KO NRK cells, treated the cells with BafA1 to prevent lysosomal degradation, lysed the cells, and immunoprecipitated the complexes using Myc-Trap Agarose beads, and immunoblotted the immunoprecipitates for CD63. We confirmed that RTN3A and CD63 are in the same complex in WT cells. The presence of RTN3A in complex with CD63 was lower in VAMP7KO cells and higher in ATG5KO cells than in WT cells (Fig. 4C, D). The most likely hypothesis explaining the presence of such a complex is that RTN3 is present in ER vesicles that exit the ER and fuse with late endosomes/MVBs in a SNARE-dependent manner. A very peculiar aspect of this result was the difference in the CD63 pattern observed by immunoblotting. CD63 is a heavily glycosylated protein that typically appears as a smear (Fig. 4C). We observed smears in the lysates, but the CD63 immunoprecipitated with Myc-RTN3A appeared as a thinner, ~50 kDa band. This band corresponds to a glycosylated form because AD1, the monoclonal antibody[46] that we use to detect rat CD63, only recognizes the native (i.e., not boiled/denatured)

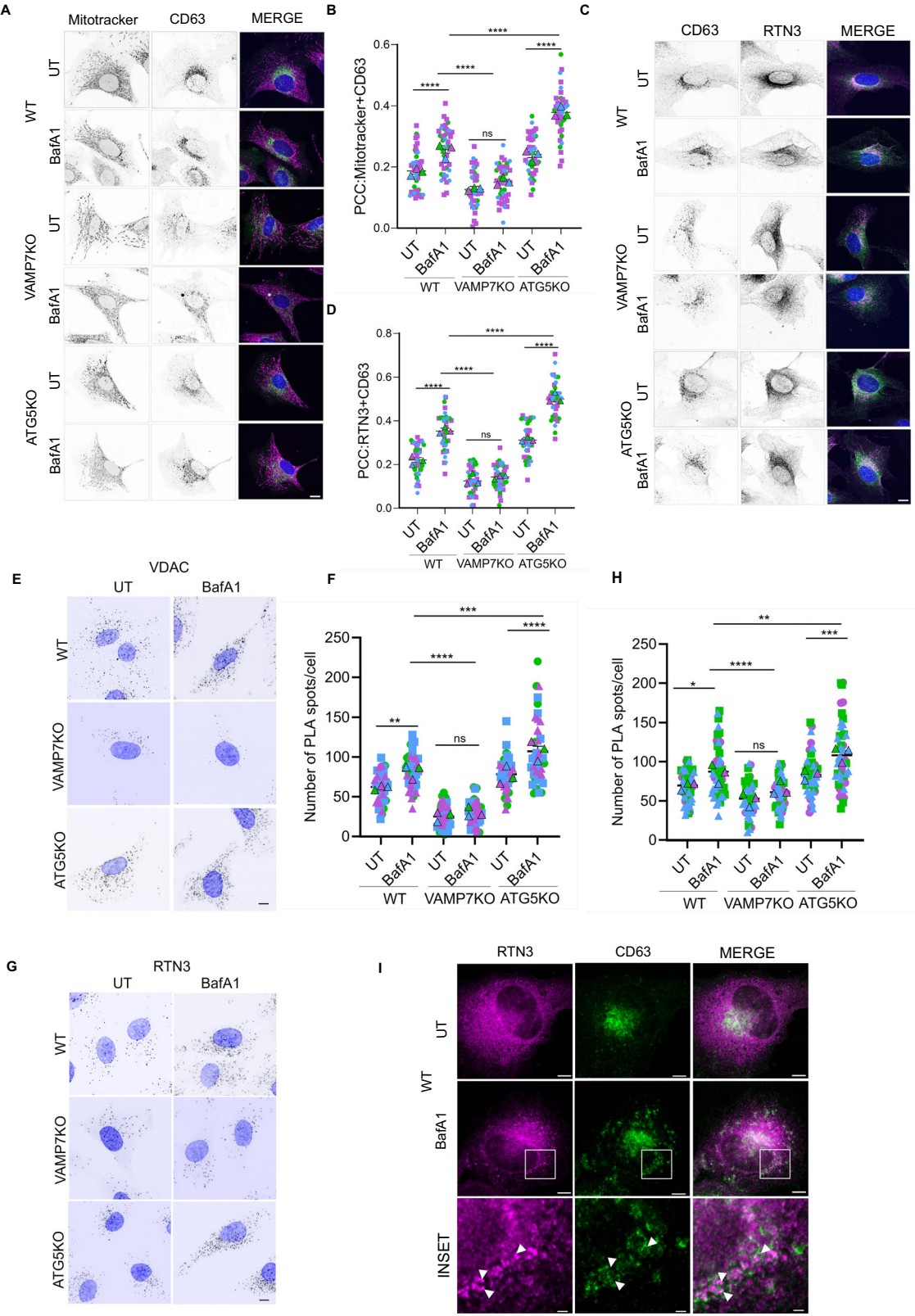

glycosylated protein (Supplementary Fig. 6B). Interestingly, the exosomal form of CD63 was also shown to have a molecular weight of ~50 kDa[47]. Therefore, the most likely hypothesis is that RTN3 interacts with glycosylated species of CD63, which concentrate in sEVs/exosomes. RTN3 and CD63 could encounter each other in late endosomes via two different routes: RTN3 via the ER to late endosomes, and CD63 via the ER to the Golgi, then to the cell surface, and finally to

endosomes. Alternatively, RTN3A could interact with an ER form of CD63 bearing the specific glycosylation pattern found here, and this complex could serve as a marker of a Golgi-bypass ER-to-late endosome route[48]. To study the state of glycosylation of CD63 when it co-immunoprecipitated with Myc-RTN3A, we co-expressed GFP-CD63, and we treated the immunoprecipitated material with Endoglycosidase H (Endo H) or PNGase F. PNGase F is expected to fully

**Fig. 3 | VAMP7 mediates transport between ER and CD63+ late endosomes.**
**A**, **B** WT, VAMP7KO and ATG5KO NRK cells were either left untreated or treated with BafA1 (100 nM) for 60 minutes and stained with Mitotracker and CD63. Scale bar = 10 μm. Pearson's Correlation Coefficient (PCC) was measured using ImageJ for the co-occurrence of Mitotracker and CD63. One-way ANOVA with Bonferroni's post hoc test was used to determine the statistical significance. Data is derived from 40 cells from three independent experiments and is presented as the mean ± SEM.****$p$-val < 0.001, ns non-significant. **C**, **D** WT, VAMP7KO and ATG5KO NRK cells were either untreated or treated with BafA1 (100 nM) for 60 minutes and stained with RTN3 and CD63. Scale bar = 10 μm. PCC was measured using ImageJ for the co-occurrence of RTN3 and CD63. One-way ANOVA with Bonferroni's post hoc test was used to determine the statistical significance. Data is derived from 40 cells from three independent experiments and is presented as the mean ± SEM.

****$p$-val < 0.001, ns non-significant. **E**–**H** WT, VAMP7KO and ATG5KO NRK cells were either left untreated or treated with BafA1 (100 nM) for 60 min and PLA was performed to quantify the RTN3/CD63 or the VDAC/CD63 interaction. One-way ANOVA with Bonferroni's post hoc test was used to determine the statistical significance. Data is derived from 50 cells from three independent experiments and is presented as the mean ± SEM. ns non-significant, *$p$-val <0.05, ** $p$-val < 0.01, ***$p$-val < 0.001. Scale bar = 10 μm. **I** WT NRK cells were either left untreated or treated with BafA1 (100 nM) for 60 minutes and immunostained for RTN3 and CD63. They were imaged using the Leica SP8 Confocal Microscope in the stimulated emission depletion (STED) module to achieve a resolution of about 50 nm. The inset shows 'annular' structures of CD63 with RTN3 spots indicated by white arrowheads in close apposition of the CD63 membrane. Scale bar = 5 μm, 1 μm.

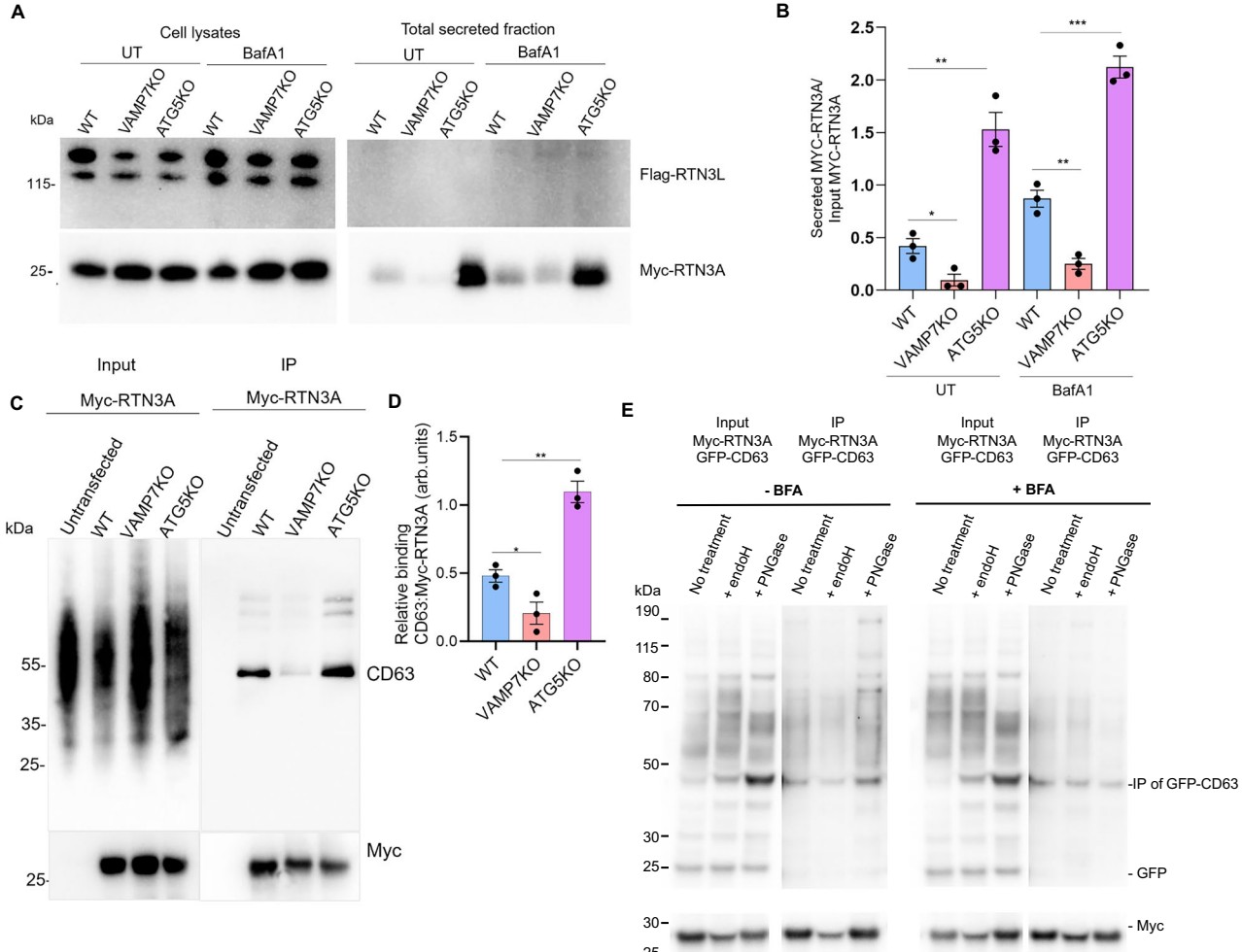

**Fig. 4 | The short form of RTN3 interacts with CD63 in a VAMP7-dependent manner. A**, **B** Secretion of the short isoform of RTN3 protein (RTN3A) is dependent on VAMP7. WT, VAMP7KO and ATG5KO NRK cells were transfected with the long Flag-RTN3L isoform and the short Myc-RTN3A isoform and either treated or not with BafA1 (100 nM) overnight in serum-free media. The total secreted fraction and cell lysates were collected and analyzed and the ratio of secreted RTN3A over total RTN3A was plotted. Unpaired Student t-test was used to determine the statistical significance. Data is derived from three independent experiments and is presented as the mean ± SEM. *$p$-val < 0.05, **$p$-val < 0.01. **C**, **D** WT, VAMP7KO and ATG5KO NRK cells were either untransfected or transfected with Myc-RTN3A, the short form of RTN3. Immunoprecipitation was performed using Myc-Trap Agarose beads and

the levels of Myc-RTN3A and CD63 were checked by immunoblotting. The relative amount of CD63 immunoprecipitated was plotted. Unpaired Student t-test was used to determine the statistical significance. Data is derived from three independent experiments and is presented as the mean ± SEM. *$p$ < 0.05, **$p$ < 0.01. **E** WT NRK cells were co-transfected with Myc-RTN3A and GFP-CD63 and treated with Brefeldin A (20 μg/ml) for 1h30. Immunoprecipitation was performed using Myc-Trap Agarose beads then beads were either left untreated or treated with Endo H or Rapid PNGase. GFP-CD63 is still present in IP even if ER-to-Golgi traffic is blocked. Cleaved GFP alone is not observed in IP demonstrating specificity of the immunoprecipitation. The experiment was replicated thrice with similar results.

deglycosylate CD63 by removing all types of N-linked glycosylation. At the same time, sensitivity to Endo H is particular for ER glycosylation pattern[49,50]. We found that both unglycosylated and glycosylated GFP-CD63 were co-immunoprecipitated with Myc-RTN3A after or not treatment with Brefeldin A, which inhibits ER-to-Golgi transport, suggesting that complex formation does not require glycosylation and passage through the Golgi apparatus (Fig. 4E). The topologies of CD63 and RTN3A are such that the glycan moiety of the luminal region of CD63 cannot interact with RTN3 which has no fully transmembrane or luminal domains. Altogether, these results provide strong evidence that ER RTN3A can be secreted following transport to late endosomes, where it forms a complex with CD63, both steps depending on VAMP7.

Work from our laboratory and others has characterized several interacting partners of VAMP7. The SNARE partners of VAMP7 include Syntaxin1, Syntaxin3, Syntaxin5, Syntaxin17, SNAP23, SNAP25, SNAP29, and SNAP47[15,16,26,51]. VAMP7 along with t-SNAREs Syntaxin1 and SNAP25 or Syntaxin3 and SNAP23 mediate the fusion of vesicles of late endosomal origin with the plasma membrane[52]. VAMP7 interacts with Syntaxin17 and SNAP29 to support autophagosome-lysosome fusion, particularly in the absence of VAMP8, and to mediate fusion of mitochondrial-derived vesicles (MDVs) with late endosomes and lysosomes[52]. We have previously described the autoinhibitory role of the VAMP7 N-terminal Longin domain[18]. To further establish the role of VAMP7 in the transport and secretion of ER components, we transfected NRK cells with the short form of RTN3 (Myc-RTN3A) as described above with or without RFP-Longin, to inhibit VAMP7 function. We observed increased co-occurrence of Myc-RTN3A and CD63 in ATG5KO cells, which was inhibited by expressing RFP-Longin, suggesting a role for VAMP7 in facilitating the transport of an ER protein to the late endosome (Fig. 5A, B).

To characterize and confirm VAMP7 partners in NRK cells, we expressed empty GFP or GFP-VAMP7 in WT NRK cells. After immunoprecipitation of GFP-VAMP7 and GFP alone and analysis of the eluates by mass spectrometry, we identified the SNARE partners of VAMP7: Rab21[53], and CD63 (Supplementary Data 3, Supplementary Fig. 6C). We considered only proteins detected in four independent experiments and excluded those present in the GFP control in each experiment. We identified proteins from different families, many of which correlate with hits found in our previously published yeast two-hybrid (Y2H) screens[15]. The pathways regulated by the interacting partners of VAMP7 unsurprisingly include membrane fusion, vesicular transport, and SNARE activity, and members possessing chloride channel inhibitor properties (Supplementary Fig. 6C, D). Of particular interest here, we found that VAMP7 coprecipitated Syntaxin17, an MDV SNARE, and Syntaxin5 and SNAP47, ER SNAREs (Fig. 5C). We have described above the VAMP7-dependent secretion of RTN3 and VDAC, which is particularly upregulated in ATG5KO cells. To decipher the mechanism of delivery of ER/mitochondrial proteins to the late endosome/MVB, we hypothesized that this occurs via vesicles originating from these organelles. Our hypothesis was supported by previous data, which suggest that VAMP7 can interact with ER (Syntaxin5 and SNAP47, Fig. 5C–E) and MDV (Syntaxin17, Fig. 5C, F) SNAREs and assist in trafficking to the late endosomal compartment[15,16,26]. We expressed GFP or GFP-VAMP7 in WT and ATG5KO cells. To inhibit VAMP7 SNARE activity, we attempted to KO VAMP7 in ATG5KO cells. We did not recover any double KO cells. We then tried to KO ATG5 in VAMP7KO, but again did not obtain any double-KO cells. Thus, we expressed the autoinhibitory Longin domain, RFP-Longin, in ATG5KO cells to acutely inhibit VAMP7. We identified the interaction of VAMP7 with ER Q-SNAREs Syntaxin5 and SNAP47 and mitochondria/autophagosomal SNARE Syntaxin17 (Fig. 5C). Interestingly, this interaction was increased in ATG5KO cells, suggesting an increase in SNARE pairing to support increased transport to late endosomes. The specificity of these interactions toward VAMP7-dependent secretion is evident from the decrease in Syntaxin5, SNAP47, and Syntaxin17 in ATG5KO cells using RFP-Longin to inhibit

VAMP7 function (Fig. 5D–F). Our results suggest that ER proteins reach late endosomes via ER-derived vesicles, which fuse with late endosomes in a VAMP7-dependent manner, depending on the ER Q-SNAREs and their t-SNARE partners, as previously shown for mitochondrial proteins and MDVs[26].

## VAMP7-mediated secretion is upregulated upon mitochondrial and ER stress

Our results so far have allowed to characterize a route of unconventional secretion which requires VAMP7 and is involved in the secretion of ER and mitochondrial proteins, with a particular reinforcement in the absence of ATG5 expression (i.e., degradative autophagy). We now aimed to understand the importance of this secretion mechanism in maintaining cellular health and survival when they are challenged with inducers of ER or mitochondrial stress. The cellular milieu inherently regulates unconventional secretion, and an uptick in cellular stress, including ER stress, can increase this secretion to help cells adapt and recover[3]. Pharmacological triggering of ER stress was shown to enhance unconventional secretion facilitated by the Golgi protein GRASP55[54]. We induced ER stress in NRK cells by treating them with Thapsigargin, a high-affinity, non-competitive SERCA inhibitor, and mitochondrial stress with Antimycin A, an inhibitor of mitochondrial respiration. We analyzed the total secreted fraction from WT, VAMP7KO, and ATG5KO cells, either untreated or treated with Thapsigargin, and observed increased secretion of RTN3 under ER stress. We also observed a similar increase in mitochondrial VDAC secretion when mitochondria were depolarized and stressed with Antimycin A (Fig. 6A–D). We found that Antimycin A only stimulated secretion of VDAC. In contrast, Thapsigargin only stimulated that of RTN3A, suggesting that specific compartmental stress induced specific transport to late endosomes and subsequent secretion (Fig. 6E–H). As described previously, we visualized the association of ER and mitochondrial components with a CD63-positive late endosomal/multivesicular body (MVB) compartment. We observed an increased association of RTN3 and Mitotracker with CD63 in the ATG5KO cells, and this co-occurrence of the two signals, as measured by Pearson's correlation coefficient, could be further enhanced by inducing ER and mitochondrial stress, respectively (Fig. 6I–L). Therefore, ER and mitochondrial stress can increase the transport of their respective organelle to late endosomes and subsequent VAMP7-dependent secretion. We conclude that ER and mitochondrial proteins can be packaged into autophagosomes, which then fuse with lysosomes, resulting in degradation and recycling, as previously demonstrated (Supplementary Fig. 6E, top). As a parallel route (Supplementary Fig. 6E bottom), the ER and mitochondrial proteins such as RTN3 and VDAC could exist most likely in ER/mitochondria-derived vesicles, which could fuse with the late endosomes/MVBs and further fuse with the plasma membrane to release the content extracellularly, all in a VAMP7-dependent manner (Supplementary Fig. 6E). This unconventional secretory mechanism would be further activated when ER and/or mitochondria are stressed, likely to prompt stress relief.

## VAMP7KO glioblastoma cells generate more necrotic tumors

Based on previous results, we reasoned that autophagy-dependent late endosomal secretion might play a crucial role in pathological stress conditions, often met in diseases such as cancer. Indeed, exosome release contributes to invasion, immune modulation, and tumor growth[55]. In addition, it has been reported that glioblastoma (GB) cells exhibit pronounced ER and mitochondrial stress, which enhances the secretion of stress-associated vesicles that modulate the tumor microenvironment[56,57]. We also realized that lower VAMP7 expression is associated with higher GB aggressiveness in various cohorts of human GB (Fig. 7A–C) and of other primary brain tumors (Supplementary Fig. 7B–F), including GB from the GBM-Mark cohort (in-house[58], Fig. 7A), or from the TCGA database (extraction of patients

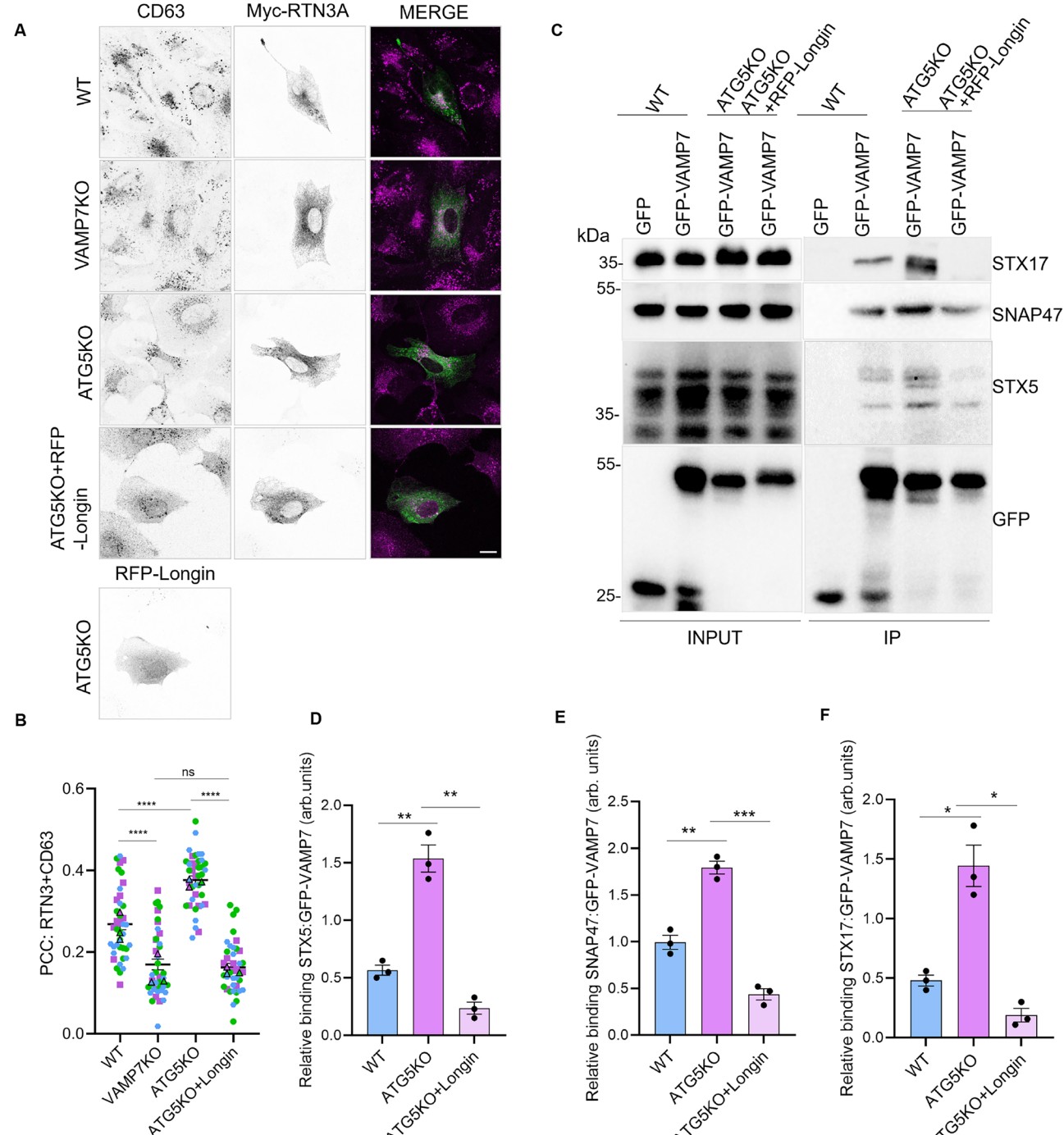

**Fig. 5 | The short form of RTN3 is transported to CD63+ late endosomes in a VAMP7-dependent manner. A** WT, VAMP7KO and ATG5KO NRK cells were transfected with either only Myc-RTN3A or a combination of Myc-RTN3A and RFP-Longin and immunostained with CD63. Scale bar = 15 μm. **B** Pearson's Correlation Coefficient (PCC) was measured for the co-occurrence of Myc-RTN3A and CD63. One-way ANOVA with Bonferroni's post hoc test was used to determine the statistical significance. Data is derived from 40 cells from three independent experiments and is presented as the mean ± SEM. ****$p$-val < 0.0001, ns non-significant.

**C** WT and ATG5KO NRK cells were transfected with either empty GFP, GFP-VAMP7 or a combination of GFP-VAMP7 and RFP-Longin. Immunoprecipitation was performed using GFP-Trap Magnetic Agarose beads and the levels of GFP-VAMP7 and partner SNAREs were checked by immunoblotting. **D–F** Relative binding of ratio of SNAP47, STX5 and STX17 to GFP-VAMP7 was analyzed and plotted. Data is derived from three independent experiments and is presented as the mean ± SEM. Welch's $t$-test was used to determine statistical significance across three independent experiments. *$p$-val < 0.05, **$p$-val < 0.01.

with Stupp treatment Fig. 7B, extraction of mesenchymal (MES) tumors, Fig. 7C). Beyond GB, low expression of VAMP7 was thus significantly associated with poorer prognosis in a large panel of primary tumors from the central nervous system, likely indicating its involvement in a general mechanism in those tumors. Interestingly, as with VAMP7, RTN3 expression negatively correlated with aggressiveness

(Supplementary Fig. 7G, H), suggesting a possible relationship between the two players in regulating GB pathogenesis. This led us to develop a model using rat RG2 cells grafted into the brain of immunocompetent F344 rats[59]. We searched for IDH1 and IDH2 mutations in RG2 cells by RT-PCR and found them to be WT (Supplementary Data 4). Therefore, the RG2 cell line could be considered a relevant

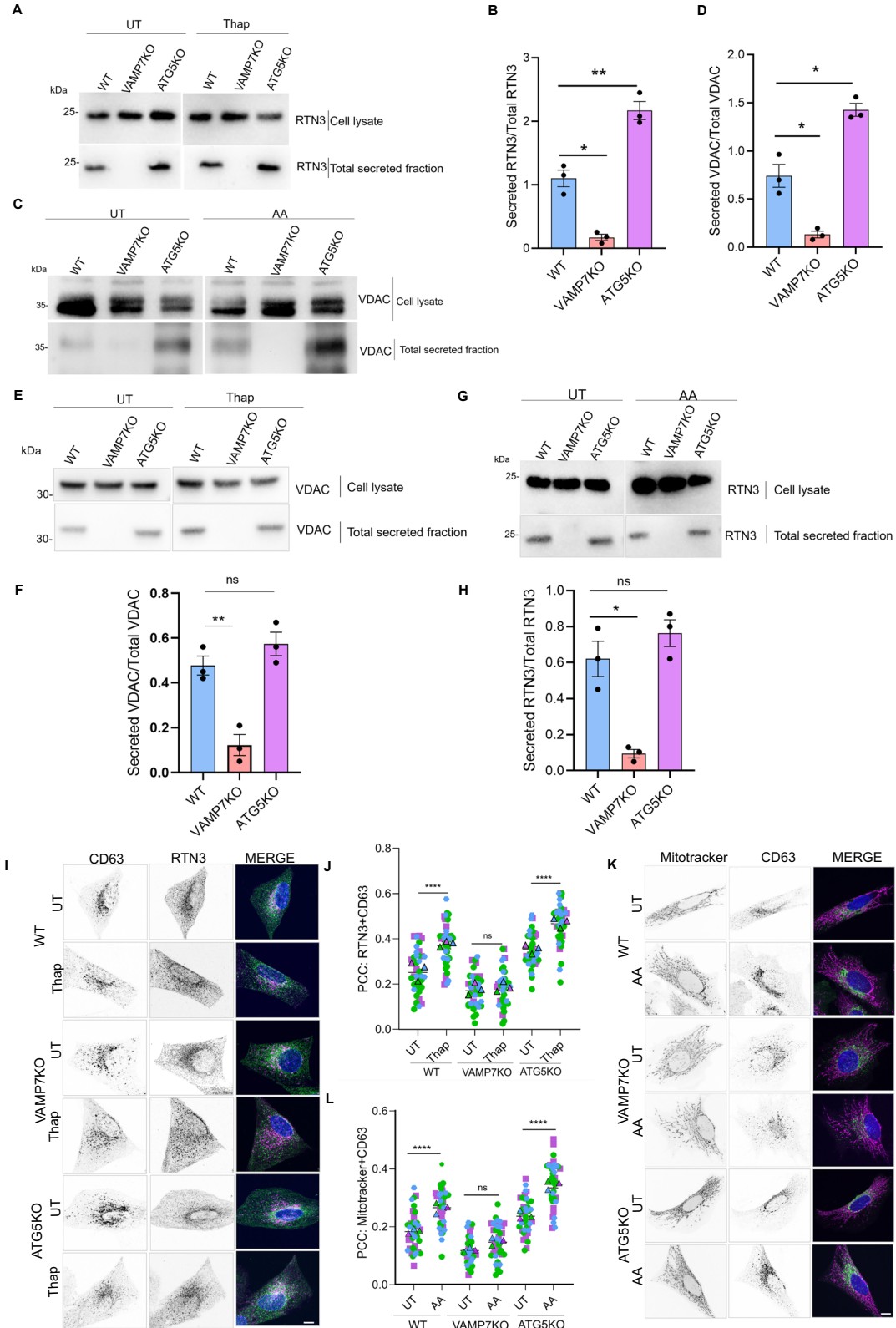

model for mimicking the most aggressive and frequent forms of adult glioblastoma, characterized by high proliferative capacity, extensive necrosis, profound metabolic rewiring, and a highly immunosuppressive microenvironment, compared with IDH-mutant glioma cells[60]. We generated VAMP7 and ATG5KO RG2 cells (Supplementary Fig. 6A) and verified that they exhibited phenotypic traits similar to those of NRK cells. Expectedly, VAMP7KO RG2 cells were impaired for the release of RTN3A and VDAC, while ATG5KO RG2 cells released more of the ER

and mitochondrial components (Fig. 7D). As for NRK cells, both KO in RG2 cells led to impaired mitochondrial basal respiration, maximal respiration, ATP production, and spare capacity respiration (Fig. 7E). We then grafted parental WT, VAMP7KO, and ATG5KO RG2 cells into the brain of F344 rats in a syngeneic model. Two weeks after injection, we either extracted the fresh tumors for direct analysis or fixed them for histological analyses. Direct analysis of fresh and fixed tumors showed that VAMP7KO cells generated considerably larger tumors

**Fig. 6 | VAMP7-mediated secretion and transport to late endosomes are upregulated due to mitochondrial and ER stress. A–H** WT, VAMP7KO, and ATG5KO NRK cells were either untreated (UT) or treated with Antimycin A (AA, 2 μM) or Thapsigargin (Thap, 100 nM) and serum starved overnight. The next day, the secreted media was processed, and total protein was precipitated using acetone. The total secreted fraction and cell lysates were collected and analyzed and the ratio of secreted RTN3A over total RTN3A (**A, B, G, H**) or secreted VDAC over total VDAC (**C, D, E, F**) was plotted. Two-tailed Welch's t-test was used to determine the statistical significance. Data is derived from three independent experiments and is presented as the mean ± SEM. \*p-val < 0.05, \*\*p-val < 0.01, ns non-significant. **I, J** WT, VAMP7KO and ATG5KO NRK cells were either left untreated or treated with Thapsigargin (Thap, 1 μM) for 60 minutes, fixed and stained with RTN3 and CD63.

Scale bar = 10 μm. Pearson's Correlation Coefficient (PCC) was measured using ImageJ for the co-occurrence of RTN3 and CD63. One-way ANOVA with Bonferroni's post test was used to determine the statistical significance. Data is derived from 40 cells from three independent experiments and is presented as the mean ± SEM. \*\*\*\*p-val < 0.0001, ns non-significant. **K, L** WT, VAMP7KO and ATG5KO NRK cells were either left untreated or treated with Antimycin (AA, 25 μM) for 45 min and stained with Mitotracker and CD63. Scale bar = 10 μm. PCC was measured using ImageJ for the co-occurrence of Mitotracker and CD63. One-way ANOVA with Bonferroni's post hoc test was used to determine the statistical significance. Data is derived from 40 cells from three independent experiments and is presented as the mean ± SEM. \*\*\*\*p-val < 0.0001, ns non-significant.

than ATG5KO cells (Fig. 7F–I). This is consistent with previous studies showing reduced growth of autophagy-deficient tumors[61,62]. VAMP7KO-derived tumors showed twice as much necrosis as WT-derived tumors and four times more than ATG5KO-derived tumors (Fig. 7J, K). Macrophages invaded much less VAMP7KO tumors than WT tumors but were equally present in the periphery of VAMP7KO compared to WT. ATG5KO tumors were less invaded by macrophages overall, including the periphery, but this might be related to their much smaller volume. In contrast, neutrophil infiltration remained throughout the tumor region in all cases (Fig. 7L–N, Supplementary Fig. 7I–J). The decreased macrophage infiltration of VAMP7KO tumors could be related to the increased necrosis, since when macrophage recruitment is impaired or delayed, necrotic areas tend to expand and persist. Altogether, our findings support the notion that late endosomal secretion is likely a determinant of macrophage recruitment to tumors. Further experiments should test the effect of RG2 secretome without grafting the cells in vivo. One might conclude from those experiments that late endosomal secretion plays an important, crucial stress-related signaling role in tumor progression by inhibiting tumor growth, most likely by releasing stress-related molecules that are detected by macrophages and enhance a tumor-suppressive microenvironment.

## Discussion

Here, we show that autophagy-dependent late endosomal secretion mediated by VAMP7 plays a crucial role in the clearance of both mitochondrial and ER components. We demonstrate that blunting this mechanism leads to pathophysiological consequences in tumors in vivo. Loss or inhibition of VAMP7 leads to alterations in mitochondrial structure and function, accompanied by clear signs of ER homeostasis disruption. In VAMP7KO cells, mitochondrial and ER-derived components fail to be delivered to late endosomes and are consequently unable to be released by the cells. In vivo, this pathway is particularly relevant under pathological stress conditions, as illustrated by VAMP7KO GB, which exhibits reduced macrophage infiltration despite increased necrosis.

On the other hand, ATG5KO cells consistently exhibit endosomal secretion-phenotypic traits opposite those of VAMP7KO cells. As we previously demonstrated in neuronal cells, we now find that ATG5KO fibroblasts and GB cells secrete greater amounts of ER and mitochondrial components, a process we consistently found to depend on VAMP7. The increased endo-lysosomal secretion observed in ATG5KO aligns well with recent findings reporting a similar phenomenon in neutrophils due to heightened lysosomal sensitivity to damage[28]. At the molecular level, we observed increased formation of SNARE complexes involving Syntaxin17, Syntaxin5, and SNAP47 in ATG5KO cells, suggesting increased fusion events engaging VAMP7 with these Q-SNAREs, thus enhancing transport from the mitochondria or the ER to late endosomes. In the ER, this membrane trafficking pathway is likely mediated by ER-derived vesicles, as we observed increased punctate RTN3 at late endosomes in ATG5KO cells and upon BafA1 treatment. These ER-derived vesicles are probably similar to the

Atlastin-1 vesicles originating from ER tubules[51]. The trafficking of such ER-derived vesicles and their fusion with late endosomes has not been previously proposed. It would differ from piecemeal micro-ER-phagy leading to degradation of ER components[44,63]. Interestingly, it mirrors the mechanism described by the Fon laboratory, in which mitochondria-derived vesicles (MDVs) fuse with late endosomes via the Q-SNAREs Syntaxin17 and SNAP29 on MDVs and the R-SNARE VAMP7 on late endosomes[26]. Accordingly, expressing the Longin domain of VAMP7 inhibited the VAMP7 complex with both Syntaxin5 and 17 as well as the transport of RTN3A and mitochondrial components to late endosomes. Therefore, we propose that the Syntaxin5/SNAP47/VAMP7 SNARE complex mediates the fusion of ER-derived vesicles while Syntaxin17/SNAP29/VAMP7 mediates the fusion of MDVs with late endosomes. Interestingly, a key study[64] used co-immunoprecipitation to show that the association between Synaptotagmin VII (Syt VII) and the tetraspanin CD63 is palmitoylation-dependent. This work demonstrated that Syt VII and CD63 form complexes that are directed to lysosomes. Earlier work from the same group had also identified VAMP7 as the R-SNARE cooperating with Syt VII in lysosomal exocytosis[17], further supporting a coordinated functional relationship among these proteins. Our observation that RTN3A and CD63 interact (Fig. 4E), together with the detection of CD63 in VAMP7 immunoprecipitates, further supports the idea that these three proteins converge within late endosomes/MVBs. Future studies should investigate how the assembly and disassembly of potentially larger VAMP7-containing complexes are regulated under different physiological conditions and disease-related states, as previously suggested[65].

VAMP7KO cells did not exhibit any autophagy block. This may be due to the involvement of other SNARE proteins, such as Ykt6p, VAMP8, and Sec22b, in various membrane fusion steps of the autophagic process in mammalian cells. Therefore, autophagy-dependent late endosomal secretion mediated by VAMP7 appears to function in parallel to degradative autophagy. Since we showed that the short form of RTN3, RTN3A, is secreted by epithelial and GB cells, and since it was reported that the long form RTN3L is not secreted but plays a role in degradative autophagy[44], we propose that the mechanism of VAMP7-dependent late endosomal secretion differs from secretory autophagy (i.e., secretion of autophagosomes), which is mediated by Sec22b[66]. RTN3A has 2 LIR motifs, whereas RTN3L has 6 LIRs. Increased secretion of RTN3A, mediated by VAMP7 (Fig. 1B–E, Fig. 4) is likely to represent a compensatory mechanism for the lack of degradative reticulophagy in ATG5KO. Therefore, the two isoforms of RTN3 could illustrate these parallel mechanisms of secretion (RTN3A) and degradation (RTN3L). The potential role of the long amino-terminal tail of RTN3L in distinguishing secretion from degradation will certainly merit further attention.

The interaction between VAMP7 and CD63 (Supplementary Data 3), which mediates cholesterol sorting into intraluminal vesicles of late endosomes[67], may play an essential role in maintaining cholesterol homeostasis. In VAMP7 knockout cells, alterations in ER homeostasis are indicated by transcriptomic signatures and the appearance of ER whorls (Supplementary Fig. 3B), features consistent

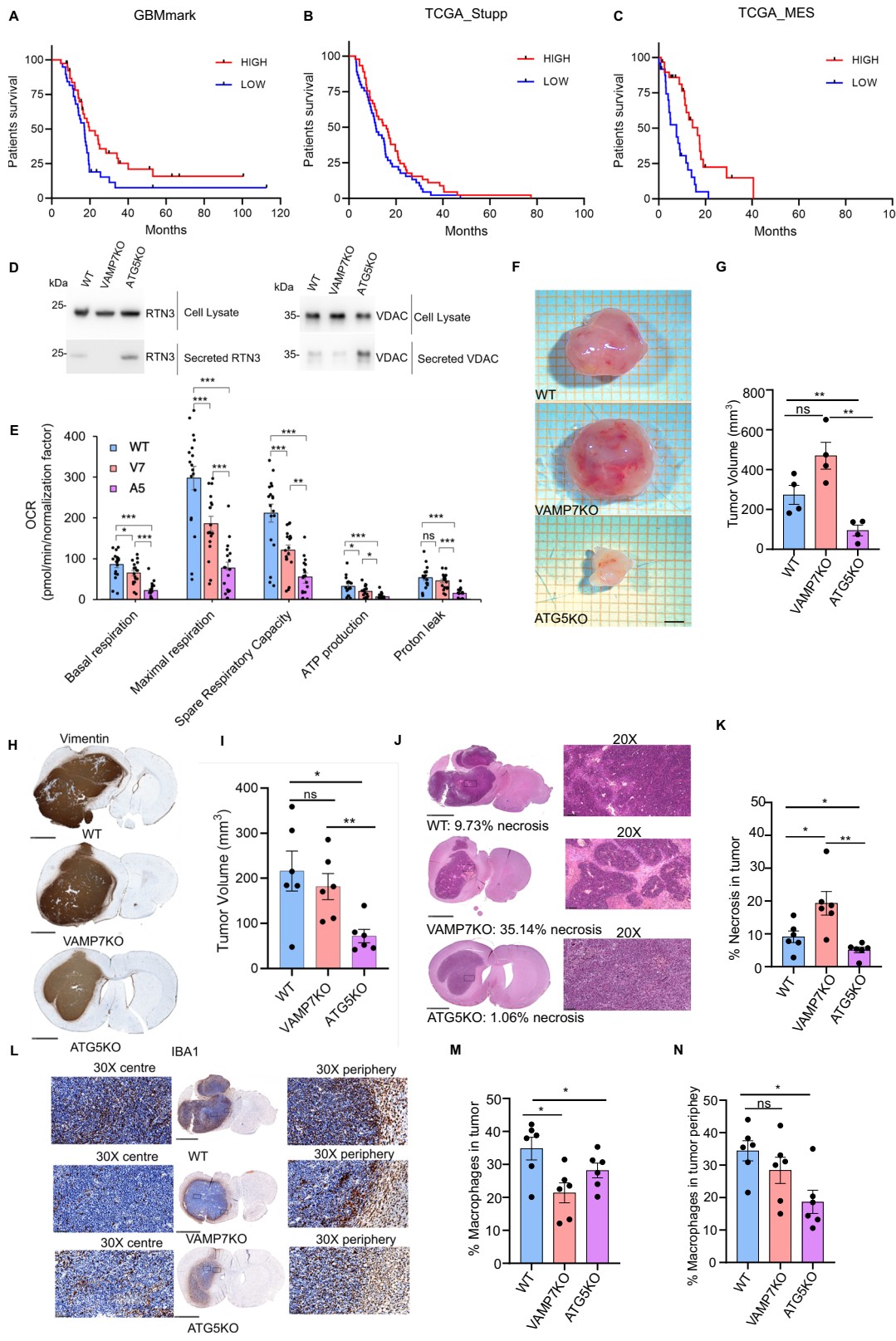

with disruptions in cholesterol metabolic pathways. These changes coincide with the coordinated activation of an ATF6-dependent cholesterogenic program[68,69] and ATF4-dependent regulation of cholesterol metabolism[70].

At the same time, VAMP7KO cells exhibit impaired mitochondrial structure and function, suggesting a convergent role in both ER and mitochondrial quality control. Our findings align with the hypothesis

that VAMP7 mediates ER-derived vesicle and MDV fusion with late endosomes, followed by exocytosis of the newly formed compartment. We also observed that further induction of ER and mitochondrial stress enhances VAMP7-dependent secretion of their respective components, suggesting that cellular stressors are essential regulators of this pathway. The observation that late endosomes secrete ER and mitochondrial membrane proteins supports the idea that these

**Fig. 7 | VAMP7KO glioblastoma cells generate more necrotic tumors.**
**A**–**C** Kaplan-Meier survival curve of patients bearing VAMP7 *low* (blue) or *high* (red) expression in the GBM-mark cohort[58], TCGA Stupp and TCGA MES (mesenchymal GB). p-val = 0.036; 0.044; 0.0005, respectively. **D** WT, VAMP7KO and ATG5KO RG2 cells were treated with BafA1 (100 nM) to inhibit degradative autophagy and serum-starved overnight. The next day, the secreted media was processed, and total secreted protein was precipitated using acetone. The total secreted fraction and cell lysates were collected and analyzed. The experiment was replicated thrice with similar results (**E**) Oxygen consumption rate (OCR) rate of basal respiration, maximal respiration, spare capacity and ATP production is significantly lower in VAMP7KO and ATG5KO RG2 cells compared with WT, indicating respiratory chain inefficiency in VAMP7KO and ATG5KO RG2 cells. Statistical significance was determined by ANOVA following by a Post Hoc group comparison with Bonferroni correction (ns: non-significant, *p-val < 0.05, **p-val < 0.01, ***p-val < 0.001, *n* = 18/group). **F**, **G** Measurement of freshly extracted tumors showed a tendency towards larger tumors in the grafting of VAMP7KO RG2 cells and significantly smaller in the grafted ATG5KO RG2 cells. Scale bar = 2 mm. Two-tailed Welch's t-test was used to determine statistical significance. Data is derived from 4 animals/genotype from three independent cohorts presented as the mean ± SEM. *p-val < 0.05; ns: non-significant. **H**, **I** IHC for Vimentin in the fixed tumors showed significant decrease in

the tumor volume of ATG5KO RG2 tumor. Scale bar = 2.5 mm. Data is derived from 6 animals/genotype from three independent cohorts presented as the mean ± SEM. **p-val < 0.01, *p-val < 0.05; ns: non-significant. Two-tailed Mann-Whitney test was used to determine statistical significance. **J**, **K** Haematoxylin-Eosin staining of the WT, VAMP7KO, and the ATG5KO RG2 tumors showed increased necrosis in the VAMP7KO RG2 tumors and a decrease in the same for the ATG5KO RG2 tumors. Scale bar = 2.5 mm. Data is derived from 6 animals/genotype from three independent cohorts presented as the mean ± SEM. **p-val < 0.01, *p-val < 0.05. Two-tailed Mann-Whitney test was used to determine statistical significance. **L** Tumor sections were double stained with HES and anti-Iba1 antibodies (macrophage and microglial cells) and revealed with secondary antibodies coupled to HRP and DAB. Magnification of the tumor center or periphery are shown for tumors derived from WT, VAMPKO or ATG5KO cells. Scale bar = 2.5 mm. **M** Quantitation of macrophage/microglia staining in the tumor center. Data is derived from 6 animals/genotype from three independent cohorts presented as the mean ± SEM.*p-val < 0.05. Two-tailed Mann-Whitney test was used to determine statistical significance. **N** Quantitation of macrophage/microglia staining at the tumor periphery. Data is derived from 6 animals/genotype from three independent cohorts presented as the mean ± SEM. *p-val < 0.05; ns non-significant. Two-tailed Mann-Whitney test was used to determine statistical significance.

---

membrane proteins are loaded into intraluminal vesicles within late endosomes and subsequently released as exosome-like vesicles. This concept is further reinforced by the identification of 'mitovesicles,' which are secreted in the brain[71] Furthermore, these results raise the possibility of an alternate mitochondrial quality control pathway. While the increase in mitochondrial mass in ATG5KO cells is explained by decreased autophagy, concomitant with decreased mitochondrial respiratory activity due to impaired quality control, VAMP7KO cells are not deficient in autophagy yet still exhibit mitochondrial defects. This suggests that perturbation of MDV trafficking might affect mitochondrial quality control in a manner that autophagy cannot resolve. Blocking conventional secretion was previously shown to increase unconventional secretion, at least in the case of CFTR, following a mechanism involving RTN3L and ER stress[72]. This might suggest crosstalks between conventional and unconventional secretions yet to be further explored. Similarly, ER stress and the loss of the Golgi-bypass ER-to-late endosome trafficking route in VAMP7KO might impact conventional secretion, as suggested by a very recent work[73], warranting further investigation. The role of exosomes in cell communication remains controversial. Our finding that VAMP7 plays an essential role in the secretion of CD63[9], is pivotal to understanding the biogenesis and function of these extracellular vesicles. Since its discovery as the product of Sybl1, an X-inactivated gene in humans[74] VAMP7 has shown original features among SNAREs. Most importantly, increased copy number was found to disrupt the human male urogenital tract via estrogen[75]. The loss of VAMP7 affects the cell sphingolipidome, with increased ceramides and GM3s, increased glycerol, and altered phosphatidyl ethanolamines[27,76]. These metabolic changes can be directly linked to ER and mitochondrial defects, and possibly to defects in exosomal secretion. The mildness of the complete KO phenotype is thus likely explained by the capacity of autophagy to at least partially compensate for the in vivo inactivation of VAMP7, except under certain conditions, such as heavy stress, which may be related to cancer. Here, we found that glioblastoma is a pathological condition that allowed us to uncover the function of autophagy-dependent late endosomal secretion mediated by VAMP7. The role of late endosomal secretion in glioblastoma, unraveled here, is most likely related to ER stress and the unfolded protein response, which are generally seen in cancer cells[77]. Interestingly, VAMP7KO tumors, although larger, exhibited increased necrosis, which may result from reduced macrophage-mediated debris removal, defective stress-adaptive secretion, and impaired lipid clearance[78]. Consistent with this, macrophages invaded less VAMP7KO tumors, whereas neutrophil

infiltration was unaffected (Fig. 7L–N, Supplementary Fig.ure 7I, J). Our findings support the notion that late endosomal secretion is a critical determinant of macrophage recruitment to tumors. The attenuation of ER- and mitochondria-derived protein secretion in VAMP7KO tumors likely reduces the release of stress-associated signals that may contribute to macrophage attraction to aggressive tumor regions. We propose a mechanism through which macrophages could detect cellular stress, a mechanism of high relevance in maintaining tissue homeostasis[78]. Since GB-associated macrophages predominantly exhibit immunosuppressive functions, it is reasonable to propose that altering VAMP7-dependent secretion could reshape the immune landscape of inherently "cold" tumors such as GB. Such reprogramming may, in turn, create a therapeutic window that enhances the efficacy of immune checkpoint inhibitors. It will be particularly relevant to test if VAMP7 mediates the presentation of mitochondria and ER peptides by MHC class I. We did not observe an opposite phenotype in ATG5KO tumors associated with macrophage invasion (Fig. 7L–N), but this might be due to decreased tumor size (Fig. 7F–I) or to differences in macrophage subtypes. Indeed, VAMP7KO tumors may limit infiltration by tumor-suppressive macrophages while retaining tumor-supportive macrophages at the periphery. Further studies will be required to clarify these differential effects and the underlying mechanisms.

In conclusion, our study demonstrates that the loss of VAMP7 significantly impairs organelle quality control, specifically affecting the ER and mitochondria and, in turn, influencing unconventional secretion processes. We have uncovered a membrane trafficking pathway that enables tubular ER components, such as RTN3A, to reach late endosomes via SNARE-dependent membrane fusion. These findings pave the way for further research into various pathological conditions where late endosomal secretion may play a crucial role[79]. Based on our results, we propose that VAMP7-dependent late endosomal secretion is an essential mechanism for organelle quality control and intercellular communication, which may be relevant to macrophage recognition of stressed cells, such as in cancer.

## Methods
### Animals
All the procedures were performed according to the European Communities Council (Directive 2010/63/EU) guidelines and were duly approved by the Ethical Committee Comité d'Éthique en matière d'Expérimentation Animale Paris Descartes (CEEA 34), and the French Ministry of Higher Education and Research (under the reference

APAFIS#43935-2023050217203669 v6). All efforts were made to minimize the number of animals and their suffering and the animals were euthanized if two limit points as established by the ethics committee were reached. Rats were housed in a conventional facility under controlled standard environmental conditions: 2–3 rats per cage with enrichment, 21 °C, under a 12 h/12 h light/dark cycle, with water and food available ad libitum.

## Stereotaxic injection of rat glioblastoma cells

All surgeries were performed in a stereotaxic frame (Kopf Instruments). 7-8-week-old male adult Fisher F344 rats were analgesized with Buprenorphine (0.1 mg/kg, Ecuphar NV - Belgique) 20 min before surgery. They were anesthetized with inhalant isoflurane (Vetflurane, Virbac, France), 3% isoflurane in an induction chamber for induction, and with 1.5–2% isoflurane in a mask into the stereotaxic apparatus for maintenance. After the onset of the anesthesia, a protective eye gel (Ocry-gel, Domes Pharma, France) was applied to the eyes to prevent drying-up during the surgery. The skin of the skull was shaved, disinfected with Vetedine solution, locally anesthetized with a Lidocaine/Prilocaine (Intervet - France) ointment and then incised to gain access to the skull. $1 \times 10^6$ of WT, VAMP7KO or ATG5KO cells suspended in 20 µL of Hank's medium were injected using a 25 µL Hamilton micro syringe (702, Hamilton) with a beveled 22S Gauge needle. A micro syringe pump with its controller (kd Scientific Legato 130) and its digital coordinate reader (Stoelting) were used to control the speed of the injection at 3 µL/min. The needle was slowly lowered to the target site and was slowly removed 5–10 min after injection was complete. Injections were targeted to the striatum (+1.0 mm AP, +3.0 mm ML, −4.0 mm DV). Surgical glue (Vetbond, SMI AG, Belgique) was applied in a thin layer to the skull around the craniotomy to prevent any deformation associated with tumor formation. The skin was then sutured with non-absorbable braided silk thread. Sterile NaCl (10 ml/kg) was injected subcutaneously to prevent dehydration after surgery.

## Post procedure monitoring

The animals were allowed to recover from the procedure before returning to their home cages, after which they were given water and food ad libitum. They were assessed clinically on a daily basis for the apparition of signs of raised intracranial pressure (lethargy, vomiting, cachexia) or focal neurological signs (hemiparesis). The subjects were weighted weekly and at the time of death. Sacrifice was carried by CO2 inhalation or an intraperitoneal injection of Pentobarbital (140 mg/kg, Euthasol 40%, Le Vet B.V, Netherlands) 14 days from the implantation procedure. Brains were retrieved immediately after sacrifice, the tumors were either excised and imaged on millimetric graph paper, or the entire brains were fixed using a 4% PFA (prepared from PFA powder, Sigma-Aldrich, 441244) and processed for immunohistochemistry.

## Immunohistochemistry (IHC) and quantification

For IHC, samples were fixed in 4% PFA for at least 72 h, embedded in paraffin at least 12 hours and sliced (4 µm) using a Leica microtome on Superfrost Plus slides (VWR, 631-0108) prior to drying at 60 °C for 1 h. The immunochemistry experiments were performed using the Discovery XT machine (Roche) and the Chromo-Map DAB kit (Roche). The following primary antibodies: Vimentin, Abcam ab92547 (EPR3776), diluted 1/250; Iba1 1/800 WAKO, 019-19741; were incubated for 1 hour at 37 °C. To perform the analysis, glass slides were digitized with the scanner Nanozoomer 2.0-RS Hamamatsu. Quantification of vimentin expression and Iba1 staining was performed using the QPATH. For quantifying tumor area and the ratio of tumor surface to specific staining for macrophages or neutrophils, we used the QPATH software. Tumors were easily delineated in the brains upon HES staining. Based on the delineation of the tumor area, the percentage of staining was quantified in all tumor sections.

## Patients' cohort and analyses

The experiments conformed to the principles set out in the WMA Declaration of Helsinki and the Department of Health and Human Services Belmont Report. All tumors were frozen after surgical resection. These tumors were either clinically or genetically characterized in the department of neurosurgery of the Pontchaillou University hospital and obtained from the processing of biological samples through the Centre de Ressources Biologiques (CRB) Santé de Rennes BB-0033-00056 (GBM-Mark cohort). The research protocol was conducted under French legal guidelines and fulfilled the requirements of the local institutional ethics committee. GBM were classified according to (i) the presence of IDH1, OLIGO2 and TP53 expression and (ii) tumor phenotype (size and form of tumor cells, hyperplasia, necrosis, proliferation index). The cohort is composed of about 100 tumors from patients with balanced age, gender, and homogeneous treatment[100] and fully documented clinical data[58,101,102]. Microarray data for these 76 tumors were analyzed[101] and expression of VAMP7 mRNA used to supervise these tumors in two groups expressing VAMP7 mRNA at low or high levels. Patient survival associated with these two groups was determined and visualized using Kaplan-Meier representation. To confirm the results obtained with the GBM-MARK cohort, we used an independent cohort from the GB TCGA dataset. These data were curated to extract exclusively patients exposed to the STUPP protocol and for which the corresponding survival data were available. We identified 226 patients with these criteria[103]. This subset of data comprises 226 patients with homogenous age range, and clinical data. The following analysis of VAMP7 mRNA expression in normalized data revealed tumors with high or low VAMP7 mRNA expression. We then evaluated the survival of patients with tumors expressing the highest (top quartile) and the lowest (bottom quartile) levels of VAMP7. In addition, public datasets for different types of primary brain tumors were analyzed for the expression of VAMP7 and RTN3 regarding survival outputs using Gliovis (https://gliovis.bioinfo.cnio.es/ - GlioVis data portal for visualization and analysis of brain tumor expression datasets[104]).

## Cell culture

Normal Rat Kidney (NRK) and Rat differentiated malignant glioma (RG2) cell lines purchased from ATCC were grown at 37 °C and 5% $CO_2$ in high-glucose DMEM (ThermoFisher, 41966029) supplemented with 10% Fetal Bovine Serum (Biosera, FB-1051H), 2mM L-Glutamine (ThermoFisher, 25030024) and Penicillin-Streptomycin (ThermoFisher, 15070063). The media was changed every 2 days, and the cells were sub-cultured 1:5-1:10 once in 4-5 days. Starvation was induced by replacing cell medium with EBSS (ThermoFisher, 24010043) supplemented with 0.2% BSA for 2 hours at 37 °C and 5% $CO_2$, followed, in the refeeding condition, by refeeding the cells with full cell medium for 30 min.

## Plasmids

Plasmids used in the study were as follows: pEGFP-C3 (Clonetech Technologies), GFP-VAMP7, RFP-Longin (described in ref. 80), pcDNA3.1-Myc-His RTN3A1 was a gift from Jimtong Horng (Addgene plasmid # 31087; http://n2t.net/addgene:31087; RRID:Addgene_31087), GFP-CD63 was a gift from John Paul Luzio (Cambridge), Flag-RTN3L was a gift from Ivan Dikic[44], ptfLC3 was a gift from Tamotsu Yoshimori (Addgene plasmid # 21074; http://n2t.net/addgene:21074; RRID:Addgene_21074).

## Antibodies and reagents

Primary and secondary antibodies used in the study were as follows:
 Anti RTN3 (Abcam, ab187764, Western blotting [WB] 1/1000, Immunoflouorescence [IF] 1/250), Anti CD63 (BD Pharmingen, 551458, WB 1/1000, IF 1/250), Anti VDAC1/2 (Proteintech, 55259-1-AP, Abcam, ab14734, WB 1/2000), Anti Myc (Cell Signaling, 2272S, WB 1/1000) Anti

Tubulin (Proteintech, 10094-1-AP, WB 1/5000), Anti GFP (Proteintech, 50430-2-AP, WB 1/4000), Anti SNAP47 (SYSY, 111403, WB 1/1000), Anti Syntaxin5 (Gift from Richard Scheller, WB 1/1000), Anti Syntaxin17 (Sigma-Aldrich, HPA001204, WB 1/1000), Anti GRP78/Bip (Proteintech, 11587-1-AP, WB 1/2000), Anti Phospho-eIF2α (Ser51) (Cell Signaling, D9G8, WB 1/1000), Total eIF2α (Cell Signaling, D7D3, WB 1/1000), GAPDH (Sigma-Aldrich, G9545, WB 1/10000), Anti p-S616 DRP1 (Cell Signaling, 4494S, WB 1/1000), Anti p-S637 DRP1 (Cell Signaling, 6319, WB 1/1000), Anti DRP1 (Cell Signaling, 8570S, WB 1/1000), Anti MFF (Cell Signaling, 84580 WB 1/1000), Anti OPA1 (BD Biosciences, 612606, WB 1/1000), Anti MFN2 (Abcam, ab124773, WB 1/1000), Anti Beta-Actin (Sigma-Aldrich, A5441, WB 1/20000), Anti pS2448-mTOR (Cell Signaling, 5536 P, WB 1/1000), Anti mTOR (Cell Signaling, 2983 P, WB 1/1000), Anti LC3B (Abcam, ab48394, WB 1/1000), Anti p62/SQSTM1 (Abcam, ab56416, WB 1/5000), Anti pS757-ULK1 (Cell Signaling, 14202 T, WB 1/1000), Anti ULK1 (Cell Signaling, 8054, WB 1/1000), Anti pT389-P70S6K (Cell Signaling, 9208S, WB 1/1000), P70S6K (Cell Signaling, 9234, WB 1/1000), Anti-rabbit IgG, HRP-linked Antibody (Cell Signaling, 7074, WB 1/10000), Anti-Mouse IgG, HRP- linked Antibody (Cell Signaling, 7076, WB 1/10000), Alexa Fluor 568 Goat Anti-Mouse (ThermoScientific, A11031, IF 1/250), Alexa Fluor 568 Donkey Anti-Mouse (ThermoScientific, A10037, IF 1/250), Alexa Fluor 488 Donkey Anti-Rabbit (ThermoScientific, A21206, IF 1/250), Vimentin (Abcam ab92547, EPR3776, IHC 1/250), Iba1 (WAKO, 019-19741, IHC 1/800).

The chemicals and reagents used in the study were: MitoTracker™ Red CMXRos (ThermoFisher, M7512), Mitotracker Deep Red™ (ThermoFisher, M22426), Mitotracker Green™ (ThermoFisher, M7514), ER Tracker Red (ThermoFisher, E34250), Phalloidin Alexa647 (ThermoFisher, A22287), Bafilomycin A1 (Cell Signaling, 54645), Antimycin A (Cell Signaling, 33357), Thapsigargin (Cell Signaling, 12758), Torin1 (Invivogen, Inh-tor1), Fluorescein isothiocyanate–dextran (dextran-FITC) (Sigma-Aldrich, FD10S-100MG), Dextran Alexa Fluor™ 647 (ThermoFisher, D22914).

### KO cell lines generation by CRISPR/Cas9 genetic engineering

To generate the KO NRK and RG2 cell lines, we used the RNA-guided Cas9 endonuclease system derived from the microbial clustered regularly interspaced short palindromic repeats (CRISPR) adaptive immune system[81]. The guide RNAs sequences were chosen with the web-based selection tool CRISPR which implements scoring algorithms based on their potential off-target and on-target DNA cleavage activity[82]. After selecting the 2–3 highest-scored guide RNAs sequences, they were cloned into a pSpCas9(BB)−2A-GFP backbone plasmid. pSpCas9(BB)−2A-GFP (PX458) was a gift from Feng Zhang (Addgene plasmid # 48138; http://n2t.net/addgene:48138; RRID: Addgene 48138). The guide RNAs have already been described in our previous publication[27]. After confirmation of the correct insertion, plasmids were purified and transfected to NRK or RG2 cells. The following day, cells were trypsinized using Trypsin-EDTA (ThermoFisher, 25300054) and detached from the culture dish, pelleted and re-suspended in PBS Ca + +/Mg + + free, 1 mM EDTA, 25 mM HEPES, 1% fetal bovine serum, penicillin and streptomycin. Cells were then sorted by a Fluorescence Activated Cell Sorter by placing one GFP-expressing cell per wheel in a 96-multi wheel dish containing complete media supplemented with Penicillin, Streptomycin and Kanamycin (Euromedex, BI-KB0286-25G). Finally, cells were amplified until final confirmation of gene knockout by protein electrophoresis and western blot.

### Protein extraction, SDS-PAGE and western blotting

Cells were extracted with a TSE lysis buffer composed of 50 mM Tris pH 7.4; 150 mM NaCl; 1% Triton X-100 (Sigma-Aldrich); 0.5 mM EDTA and 1X complete protease inhibitor cocktail (Sigma Aldrich, 05056489001). After extraction, samples were centrifuged at 16,000 g at 4 °C for 20 minutes and the supernatant was recovered. Protein content was estimated using Bradford assay. For the blots involving mitochondrial proteins, the cells were lysed in RIPA buffer (50 mM Tris pH 7.4; 150 mM NaCl; 5 mM EGTA; 1% Triton X-100; 0.1% SDS; 0.5% Deoxycholate), centrifuged 10,000 g at 4 °C for 10 min. Protein estimation was performed on the cleared lysates using BCA assay. 20–30 μg of total protein was loaded into 12% Laemmli-SDS-PAGE gels and run at 80 V for 30 minutes and then 110 V for another 90 minutes approximately. Proteins were then transferred to a 0.45 μm nitrocellulose membrane at 100 V for 90 minutes at 4 °C. Membranes were blocked with a phosphate-buffer saline solution (PBS) supplemented with 5% non-fat dry milk. Primary and secondary antibodies were incubated in PBS supplemented with 5% non-fat dry milk solution. Secondary antibodies were conjugated with Horse Radish Peroxidase and the membranes were developed with SuperSignal™ West Femto Maximum Sensitivity Substrate (ThermoFisher, 34095). Images were acquired in a Chemidoc imagerTM (BioRad) using an Auto setting followed by a manual exposure for low and high exposure images as needed. Non-saturated images were analyzed with Fiji/ImageJ (National Institute of Health, NIH) to determine signal density.

### Immunoprecipitation

WT, VAMP7KO and ATG5KO NRK cells were plated on 10 cm dishes at about 60% confluency (~6 × 10^6 cells) and were allowed to attach overnight. The next day, the cells were either left untransfected or transfected with the appropriate plasmid using JetPrime transfection reagent (PolyPlus, 101000046). We transfected the cells following the manufacturer's protocol and changed the media six hours after the transfection. Thirty six hours after transfection, the cells were trypsinized using Trypsin-EDTA and processed for immunoprecipitation. We used either the GFP-Trap Magnetic Agarose beads (Chromotek, GTMA-20) or the Myc-Trap Agarose beads (Chromotek, YTA-20) for immunoprecipitation.

### GFP-Trap Magnetic Agarose Immunoprecipitation

WT and ATG5KO NRK cells were transfected with empty GFP vector, GFP-VAMP7 or a combination of GFP-VAMP7 and RFP-Longin were washed with D-PBS (ThermoFisher, 14190169), trypsinized using Trypsin-EDTA and centrifuged at 100 g for 4 minutes. The pellets were washed again with D-PBS, the supernatant was removed, and the pellets were lysed on ice for 30 min using an ice-cold lysis buffer composing of 10 mM Tris-HCl pH 7.5, 150 mM NaCl, 0.5 mM EDTA (Sigma-Aldrich), 0.5% NP-40 and 1X complete protease inhibitor cocktail. The lysates were centrifuged at 16,000 g for 20 min at 4 °C for clearing. The supernatants after clearing were collected and transferred to a new tube and protein estimation was performed using Bradford assay. The GFP-Trap beads were equilibrated by carefully washing three times with ice-cold Dilution buffer (10 mM Tris-HCl pH 7.5, 150 mM NaCl, 0.5 mM EDTA) and were incubated with 1 mg cell lysate at 4 °C for 2 h with end-to-end rotation. The unbound fraction was removed, and the beads were washed five times with ice-cold washing buffer (10 mM Tris-HCl pH 7.5, 150 mM NaCl, 0.5 mM EDTA, 0.1% NP-40) to remove non-specifically bound proteins. The bound fraction was eluted in 2X Laemmli sample buffer by boiling at 95 °C and analyzed by SDS-PAGE.

### Myc-Trap Agarose Immunoprecipitation

WT, VAMP7KO and ATG5KO NRK cells were either left untransfected or transfected with the Myc-RTN3A plasmid. The lysis and immunoprecipitation protocol were identical as for the GFP-Trap Magnetic Agarose beads. Before lysis, NRK cells were treated with BafA1 (100 nM) to inhibit lysosomal degradation. To immunoblot for CD63, the bound fraction was eluted in 2X Laemmli buffer without DTT and heated at 37 °C for 10 min before analysis by SDS-PAGE.

WT NRK cells co-transfected with the Myc-RTN3A and GFP-CD63 plasmids were also used for Myc-Trap Agarose Immunoprecipitation. The protocol was identical to that cited above.

## Deglycosylation and ER-to-Golgi transport inhibition of CD63 with Endo H/PNGase and with Brefeldin A

After Myc-Trap Agarose Immunoprecipitation of lysates from WT NRK cells co-transfected with the Myc-RTN3A and GFP-CD63 plasmids, beads were treated with Endo H (New England Biolabs, P0702L) or Rapid PNGase F (New England Biolabs, P0710S) following the protocols of the manufacturer. For Endo H treatment protocol, Endo H treatment (+Endo H) or no Endo H treatment condition (-Endo H) were preceded by denaturation step of 10-minute boiling at 100 °C with buffer containing SDS and DTT. Then, beads were incubated with Endo H at 37 °C for 1 hour 30 minutes. For Rapid PNGase F treatment protocol, beads were incubated with Rapid PNGase F at 50 °C for 10 min. Final elution of bound fraction in 2X Laemmli buffer with or without DTT, at 95 °C for 5 min or at 37 °C for 10 min respectively, was realized before being analyzed by SDS-PAGE.

To inhibit ER-to-Golgi transport of CD63, Brefeldin A (BFA, Sigma-Aldrich, B7651) was used to treat WT NRK cells co-transfected with the Myc-RTN3A and GFP-CD63 plasmids at a concentration of 20 μg/ml for 1 h 30 min before lysis. BafA1 (100 nM) was added in the cell culture media for the last 45 min of BFA treatment to inhibit lysosomal degradation.

## Exosome extraction and analysis

WT, VAMP7KO and ATG5KO NRK cells were plated on 15 cm dishes at about 80% confluency and were allowed to attach overnight. The day following cell seeding, the complete culture media was changed to serum-free DMEM with BafA1 (100 nM) and left overnight. The exosome preparation was done using the Invitrogen Total Exosome Isolation Reagent (from cell culture media) (#4478359). The cell culture media were recovered and centrifuged at 2,000 g for 30 minutes to remove cell debris. The supernatant was carefully transferred to a fresh Falcon tube and 0.5X volume of the exosome reagent was added and mixed. The tubes were incubated overnight at 4 °C, followed by a 10,000 g centrifugation for 60 minutes. In order to remove the media, the supernatant was carefully removed and 5 ml cold PBS was added, followed by a repeat centrifugation at 10,000 g for 30 min. The supernatant was removed, and the pellet was carefully mixed in 75 μl cold PBS and stored at 4 °C before analysis either by SDS-PAGE and western blotting or by Differential Light Scattering (DLS) Zeta-Sizer (Malvern Instruments).

## Secreted fraction preparation

WT, VAMP7KO and ATG5KO NRK cells were plated on 15 cm dishes at about 80% confluency and were allowed to attach overnight. The day following cell seeding, the complete culture media was changed to serum-free DMEM with or without treatment and left overnight. Cells were processed for western blot as described. The cell culture media was recovered and centrifuged at 2,000 g for 10 minutes to remove cell debris. The supernatant was then treated with 3X volume of acetone and centrifuged at 15,000 g for 30 min to recover all secreted proteins. The pellet was then re-suspended in TSE lysis buffer with Triton X-100 at a final concentration of 1% followed by SDS-PAGE and western blotting.

## Proteomics for the immunoprecipitated samples to characterize VAMP7 partners

**Protein digestion.** S-Trap™ micro spin column (Protifi, Hutington, USA) digestion was performed on IP eluates in 2X Laemmli buffer according to manufacturer's instructions. Briefly, samples were reduced with 20 mM TCEP and alkylated with 50 mM chloracetamide (CAA) for 5 minutes at 95 °C. Aqueous phosphoric acid was then added to a final concentration of 2.5% following by the addition of S-Trap binding buffer (90% aqueous methanol, 100 mM TEAB, pH 7.1). Mixtures were then loaded on S-Trap columns. Four additional washing steps were performed to ensure thorough SDS removal. Samples were

digested with 1.5 μg of trypsin (Promega) at 47 °C for 2 h. After elution, peptides were vacuum dried and resuspended in 15 μl of 2% ACN, 0.1% formic acid in HPLC-grade water prior to MS analysis.

## NanoLC-MS/MS protein identification and quantification

The tryptic peptides (1/4 of digest) were injected on a nanoelute (Bruker Daltonics, Germany) HPLC (high-performance liquid chromatography) system coupled to a timsTOF Pro (Bruker Daltonics, Germany) mass spectrometer. HPLC separation (Solvent A: 0.1% formic acid in water; Solvent B: 0.1% formic acid in acetonitrile) was carried out at 400nL/min using a packed emitter column (C18, 25 cm × 75 μm × 1.6 μm) (Ion Optics, Australia) using a 15 min gradient elution (2–17% solvent B during 8 min; 17–25% during 2 min; 25% to 37% during 2 min; 37% to 95% for 1 min and finally 95% for 2 min to wash the column). Mass-spectrometric data were acquired using the parallel accumulation serial fragmentation (PASEF) acquisition method in DIA mode. The measurements were carried out over the m/z range from 475 to 1000 Th. The range of mobility values was from 0.8 to 1.27 V s/cm$^2$(1/k0). The total cycle time was set to 0.95 s, and the number of PASEF MS/MS scans was set to 10.

## Data Processing Following LC-MS/MS acquisition

Data analysis was performed using DIA-NN software (version 1.8.1)[83]. A search against the UniProtKB/Swiss-Prot Rattus norvegicus database mixed with its trEMBL entries (updated February 2021, 36188 entries) was performed using library free workflow. For this purpose, "FASTA digest for library free search/library generation" and "Deep learning spectra, RTs and IMs prediction" options were checked for precursor ion generation. A maximum of 1 trypsin missed cleavage was allowed, and the maximum variable modification was set to 2. Carbamidomethylation (Cys) was set as the fixed modification, whereas protein N-terminal methionine excision, methionine oxidation and N-terminal acetylation were set as variable modifications. The peptide length range was set to 7–30 amino acids, precursor charge range 2–4, precursor m/z range 300–1300, and fragment ion m/z range 300–1300. To search the parent mass and fragment ions, accuracy was set to 10 ppm. The false discovery rates (FDRs) at the protein and peptide level were set to 1%. Match between runs was allowed. For the quantification strategy, Robust LC (high precision) was used as advised in the software documentation, whereas default settings were kept for the other algorithm parameters.

Statistical and bioinformatic analysis were performed with Perseus software (version 1.6.15)[84] freely available at www.perseus-framework.org. All protein intensities were log2 transformed to perform statistics. For statistical comparison, we set 2 groups, each containing up to 4 biological replicates: GFP (negative control) and VAMP7 immunoprecipitates. For the VAMP7 group, we normalized all the measured values on the VAMP7 intensity median. We then filtered the data to keep only proteins with at least 3 out of 4 valid values in at least one group. Next, the data were imputed to fill missing data points by creating a Gaussian distribution of random numbers with a standard deviation of 33% relative to the standard deviation of the measured values and 1.8 standard deviation downshift of the mean to simulate the distribution of low signal values. We performed a two-sample t-test with FDR < 0.05 and S0 = 1. The mass spectrometry proteomics data have been deposited to the ProteomeExchange Consortium via the PRIDE partner repository with the dataset identifier PXD057152[85].

## Cellular oxygen consumption (respirometry)

Mitochondrial respiration was measured, as described previously[86]. Briefly, cells were plated in Seahorse XF96 plates and allowed to adhere overnight. The following day, cell culture medium was replaced with fresh DMEM (D5030, Sigma-Aldrich) supplemented with 5 mM HEPES (pH 7.4) without FBS, and cells were allowed to stabilize for 1 h in a non-CO$_2$ incubator at 37 °C. Oxygen consumption rate (OCR) was

then quantified using the MitoStress Test and Seahorse XF96 Extracellular Flux Analyzer (Agilent) according to the manufacturer's instructions. During the test, Oligomycin (2 μM), BAM-15 (2 μM), and a mix of Rotenone/Antimycin A (2 μM) were injected sequentially to assess the different mitochondrial respiration parameters.

OCR (respiration) of RG2 lines (WT, VAMP7KO and ATG5KO) was measured by a Seahorse XF analyzer (XFp, Proteigene/Agilent). Key parameters of mitochondria functions which are basal respiration, maximal respiration, ATP production, spare capacity and proton leak, were obtained by sequential injection of Oligomycin (1 μM), FCCP (2 μM) and Rotenone/antimycin A (0.5 μM), following the standard protocol of the XFp Cell Mito Stress Kit (3 measures par phase). For each of the 3 independent experiments, cells were plated in 3 utility plates (2 replicated par cell lines and plate, $N = 18$).

## Measurement of pH and lysosomal positioning

Cells were incubated concomitantly with 250 μg/ml dextran-FITC (pH sensitive – less fluorescent in acidic pH) and 50 μg/ml dextran-AlexaFluor 647 (pH insensitive) for 16 hours in full medium, followed by a washout of 6 h and then incubation with 5 μg/ml Hoechst for 10 min (adapted from refs. 30,87,88). After, cells were live-imaged in a confocal Zeiss LSM 980 with Airyscan 2 (63X, 1.4 NA, oil immersion objective) at 37 °C. As an indication of relative pH levels, and therefore of degradative capacity, the ratio between the mean fluorescent intensities of dextran-FITC and dextran-AlexaFluor 647 were obtained for each individual lysosomal particle. For the definition of the lysosomal particles as Regions of Interest (ROIs), the channel corresponding to the signal for dextran-AlexaFluor 647 was optimized and thresholded using the adaptive thresholding tool available in the plugin 'Mitochondria Analyzer', followed by the analyze particle macro from ImageJ/Fiji. The mean fluorescent intensities for each channel were then measured with the obtained ROIs. The centroids for each ROI were also obtained and their distance relative to both the plasma membrane and the nuclear membranes (in the shortest straight line) was measured in the Python Integrated Development Environment (IDE) Spyder software. The distance of each ROI to the nucleus versus total distance (nuclear membrane-lysosome + lysosome-plasma membrane) was then determined as a ratio and each lysosomal particle was attributed to a quartile representing distance from the nucleus (0.25, 0.50, 0.75, 1). The dextran-FITC/AlexaFluor 647 ratio was then averaged across all lysosomes in each quartile for each cell.

## Flow cytometry

Plated cells were incubated with 200 nM MitoTracker GreenTM (or vehicle) dissolved in full cell medium for 30 minutes, followed by three washings with PBS. Cells were then collected by trypsinization and centrifugation and resuspended in PBS + 0.2% BSA, followed by flow cytometry-driven detection of fluorescence using a BD FACSCaliburTM (BD Biosciences).

## Immunofluorescence

WT, VAMP7KO and ATG5KO cells were plated on glass coverslips and allowed to attach overnight. Depending on the experiment, the cells were transfected or were processed directly for endogenous staining. Cells were fixed with a 4% formaldehyde solution (PFA, prepared from 32% PFA, Electron Microscopy Sciences, 15714S) for 20 min, followed by three PBS washes and permeabilization with 0.1% PBS-Triton for 5 min. After three washes with PBS, the blocking step was performed with 0.25% PBS-gelatin at room temperature for one hour. The primary and secondary antibody were prepared in 0.125% PBS-gelatin. The incubation with the primary antibodies was performed overnight at 4 °C, followed by three washes and an incubation in secondary antibody at room temperature for 1 h. The coverslips were mounted on the slides using the Vectashield mounting medium (Vector Labs, H-1000) and were immobilized using clear nail polish.

## Proximity ligation assays (PLA)

WT, VAMP7KO and ATG5KO cells were plated on glass coverslips and allowed to attach overnight. PLA was performed using the DUOLINK Kit (Sigma-Aldrich, DUO92002/04) following the manufacturer's protocol.

## Confocal microscopy

For high-magnification analysis, Z-stacks of confocal images were acquired in Zeiss LSM 880 Confocal Laser Scanning Microscope (63X, 1.4 NA, oil immersion objective), the Leica True Confocal Scanner (TCS) SP5 (63X, 1.4 NA, oil immersion objective) or the Zeiss LSM 710 Confocal Laser Scanning Microscope. In all cases, lasers and spectral bands were chosen to maximize signal recovery while avoiding signal bleed through. Live imaging was performed at 37 °C using Zeiss LSM 880 Confocal Laser Scanning Microscope (63X, 1.4 NA, oil immersion objective).

## STED microscopy

STED images were acquired within NeurImag facility using confocal laser scanning microscope LEICA SP8 STED 3DX equipped with a 93×/ 1.3 NA glycerol immersion objective and 3 hybrid detectors (HyDs). The specimens were excited with a pulsed white-light laser (598 nm or 640 nm) and depleted with a pulsed 775 nm depletion laser to acquire nanoscale imaging using SMD HyD detector. Typically, STED Images (1200 × 1200 px) were averaged 16 times in line and acquired with a magnification zoom >2 leading to a pixel size in the range of 30–35 nm. Excitation laser was adjusted to avoid any saturating pixels, and the same laser intensity and HyD sensitivity were used for both control and treated cells.

## Image Analysis

All quantifications were done using the Fiji/ImageJ or Icy software[89,90]. The 'Analyze-Measure' plugin in Fiji/ImageJ was used for intensity measurement after background subtraction and for length measurement after setting an appropriate scale. The 'Cell Counter' plugin was used for counting the mitochondria number. The Fiji/ImageJ plugin 'Mitochondria Analyzer' was used to count mitochondrial number and determine mitochondrial morphological and connectivity parameters in deconvolved high-resolution Z-stack images (deconvolution performed with the 'PSF Generator' and 'DeconvolutionLab2' plugins)[42].

For co-localization analysis, Z-stacks and unprocessed images were used for quantifications using green (Ex. 488 nm) and red (Ex. 594 nm) or far-red (Ex. 647 nm) channels. The colocalization analysis was performed using the Just Another Colocalization Plugin (JACoP) in the Fiji/ImageJ software[91]. The extent of colocalization was assessed using Pearson's correlation coefficient (PCC).

We used a custom ImageJ/Fiji script to batch-process multichannel CZI stacks and generate radial profiles around nuclei. A z-projection of the RTN3 channel was used as a proxy for the cell envelope to create a cell mask. CLAHE contrast enhancement and automatic thresholding were applied. An assisted-manual correction procedure was performed when required. Nuclei were segmented from the nuclear (DAPI) channel by projection, thresholding, size filtering and optional watershed. A Euclidean distance map of the cell mask was then computed to obtain per-pixel distances to the cell border. This map was used to determine the nucleus-to-periphery distance for each cell and to guide annulus construction. Concentric annuli were built by iterative ROI enlargement around each nucleus and Boolean-intersected with the mask, with annulus thickness set in pixels (or as a fraction of the nucleus–periphery distance) and radial positions normalized to the cell periphery so profiles are comparable between cells. For each annulus we measured area, normalized intensity and intensity density on the signal channel. Distances were normalized to the cell periphery. Per-cell per-annulus measurements,

the ROI archive and representative TIFFs (EDM pattern and summed signal) were exported as tables and images.

## Conventional Electron Microscopy

WT and VAMP7KO NRK cells were grown on glass coverslips and processed as described previously[92]. Briefly, cells were fixed with 2.5% Glutaraldehyde in 0.1 M Cacodylate buffer overnight at 4 °C, post-fixed with 1% OsO4/ 1.5% Potassium ferricyanide 45 min at 4 °C, dehydrated in ethanol and embedded in Epon resin. The samples were processed for ultramicrotomy and ultrathin sections (60–70 nm), post-stained with uranyl acetate and lead citrate and analyzed with a Transmission Electron Microscope (Tecnai Spirit G2; ThermoFischer, Eindhoven, The Netherlands) equipped with a 4k CCD camera (Quemesa, EMSIS, Münster, Germany).

## RNA-seq, data visualization and enrichment analysis

RNA was isolated from one 10 cm dish each of WT, VAMP7KO and ATG5KO NRK using the RNA-easy Midi Kit (Qiagen, 75144). Four independent replicates were used for this analysis. RNA-Seq libraries were generated from 600 ng of total RNA using Illumina Stranded mRNA Prep, Ligation kit and IDT for Illumina RNA UD Indexes Ligation (Illumina) according to manufacturer's instructions (Reference Guide - PN 1000000124518: v3). Libraries were checked for quality and quantified using Bioanalyzer 2100 (Agilent). Libraries were sequenced on an Illumina NextSeq 2000 sequencer as 50 bases reads. Image analysis and base calling were performed using RTA version 2.7.7 and BCL Convert version 3.8.4 (Illumina). Reads were preprocessed using cutadapt[93] version 4.2 to remove adapter, polyA and low-quality sequences (Phred quality score below 20), reads shorter than 40 bases were discarded for further analysis. Reads mapping to rRNA were also discarded (this mapping was performed using bowtie[94] version 2.2.8). Reads were then mapped onto the mRatBN7.2 assembly of Rattus norvegicus genome using STAR[95] version 2.7.10b. Gene expression was quantified using htseq-count[96] version 0.13.5 and gene annotations from Ensembl release 110. Differential gene expression analysis was performed using R 4.1.1 and DESeq2[97] 1.34.0 Bioconductor library, adjusting for batch effect. Gene sets were tested for enrichment using the Gene Set Enrichment Analysis (GSEA)[34] either with the Gene Ontology gene set or the Hallmark gene sets. Normalized Enrichment Score (NES) and p-values were computed with GSEA. Volcano plots and bubble plots were generated using freely available visualization and analysis tools, iDEP[98] and SRplot[99].

## Statistics and Reproducibility

All experiments were performed in the laboratory with a minimum of three independent biological replicates. Data are expressed as mean ± SEM until otherwise stated. Statistical significance between groups was determined using one-way ANOVA, Kruskal-Wallis test, mixed-model two-way ANOVA, Mann-Whitney test or Unpaired/Welch's t-test. Statistical analysis was performed using GraphPad Prism v10.6.1. No statistical method was used to predetermine sample size. We did not exclude any data from the analyses. The experiments were not randomized and required intermediate intervention, hence the investigators could not be blinded to allocation during experiments and outcome assessment.

## Reporting summary

Further information on research design is available in the Nature Portfolio Reporting Summary linked to this article.

## Data availability

All data associated with this study can be found in the paper, the Supplementary materials, and the Source data file. Research materials are available upon request. The mass spectrometry proteomics data have been deposited to the ProteomeXchange Consortium via the PRIDE partner repository with the dataset identifier PXD057152. The RNA sequencing data have been deposited to the GEO repository with the dataset identifier GSE280209 https://www.ncbi.nlm.nih.gov/geo/query/acc.cgi?acc=GSE280209. The script for the Sphere plugin has been deposited in Zenodo under the following https://doi.org/10.5281/zenodo.17473585. Source data are provided with this paper.

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

## Acknowledgements

We thank all members of the Galli team for their assistance and discussions. Work in our group was funded by grants from the French National Research Agency (MetDePaDi ANR-16-CE16-0012, GlioUPS ANR-23-CE16-0035-01), Institut National du Cancer (PLBIO 2018-149), Fondation pour la Recherche Médicale (FRM) Labellisation to TG and postdoctoral fellowship (SPF202110014120) to SV, Fondation de France

(grant #00096652), FRM grant (MND202310019903), and Fondation Bettencourt Schueller (Coup d'Elan) to TG. We would like to thank the NeurImag Imaging core Facility team (part of IPNP, Inserm U. 1266 and Université Paris Cité) and member of the national infrastructure France-BioImaging supported by the French National Research Agency (ANR-10-INBS-04) for their technical and scientific support. Our lab is part of the DIM C-BRAINS, funded by the Conseil Régional d'Ile-de-France. NR acknowledges Four Diamonds Research Foundation, FCT 2022.09311.PTDC, 2022.04407.PTDC, FCT ERC-Portugal. This project also received funding from the European Union's Horizon 2020 research and innovation programme under grant agreement MIA-Portugal No 857524 and the Comissão de Coordenação e Desenvolvimento Regional do Centro - CCDRC through the Centro2020 Programme. The content is solely the responsibility of the authors and does not necessarily represent the official views of the National Institutes of Health. We thank the Leducq establishment for funding the Leica SP8 Confocal/STED 3DX system and Sésame Région Ile-de-France for funding the Zeiss 880 Confocal/Airyscan system. The Seahorse XF analyzer was funded by the DIM Cerveau et Pensée 2015 (Neuroflux) grant to DZ. We thank the BIOSIT H2P2 platform for immunohistochemistry, in particular Gevorg Ghukasyan (https://biosit.univ-rennes1.fr/). Sequencing was performed by the GenomEast platform, a member of the 'France Génomique' consortium (ANR-10-INBS-0009). This work was supported by the Agence Nationale de la Recherche ANR "MOBIDIC" (ANR-23-CE14-0041-02) to C.D. This project was also supported by funds from the Fondation pour la Recherche Médicale (FRM, EQU202403018041 and MAT202211016240), INCa PLBio2020, Olicogcyte Bretagne, INSERM (International Research Project – TUPRIC) to EC. TG and NR received the Mariano Gago Prize from the French Academy of Sciences for the project that led to this publication. We would like to extend special thanks to Sébastien Nola for contributing the IDH1/2 data. Illustrations were created with BioRender.

## Author contributions

T.G. and S.V. designed the study and prepared the original draft of the manuscript. S.V. performed and analyzed most of the experiments. P.D. performed and analyzed mitochondrial and autophagy experiments in NRK cells. Q.L. assisted with the manuscript revision and conducted studies on the impact of glycosylation on CD63 interactions. R.P. performed all the IHC for the tumors. L.T. performed the injections into the rat brains and carried out post-procedure care. J.L. and I.C.G. performed and analyzed the proteomic studies. B.C. assisted in the animal experiments. J.B.M. and D.Z. carried out and analyzed the respirometry studies in RG2 cells. J.W. generated the KOs in the N.R.K.; C.K. analyzed the RNA sequencing data. PB developed a plugin for analyzing CD63 distribution. S.F. and N.D. assisted PD in data analysis. L.D. performed the super-resolution STED imaging. C.D. performed the EM imaging and assisted in the EM data analysis. E.C. performed the human cohort analysis, supervised the animal experiments and their data analysis, and edited the manuscript. N.R. supervised the mitochondrial studies, edited the manuscript, and generated funding. T.G. conceptualized the study, managed the project, supervised the data analysis and writing of the manuscript, and generated funding. All authors reviewed the results and approved the manuscript.

## Competing interests

The authors declare no competing interests.

## Additional information

[1]Université Paris Cité, Institute of Psychiatry and Neuroscience of Paris, INSERM U1266, Membrane Traffic in Healthy & Diseased Brain, Paris, France. [2]Multidisciplinary Institute of Ageing (MIA), University of Coimbra, Coimbra, Portugal. [3]INSERM U1242, Université de Rennes, Centre de Lutte Contre le Cancer Eugène Marquis, Rennes, France. [4]Université Paris Cité, Institute of Psychiatry and Neuroscience of Paris, INSERM U1266, PhenoBrain, Paris, France. [5]Necker Proteomics, Université Paris Cité - Structure Fédérative de Recherche Necker, INSERM US24/CNRS UAR3633, Paris, France. [6]IGBMC, CNRS UMR 7104, Inserm U1258, Université de Strasbourg, Illkirch, France. [7]Université Paris Cité, Institute of Psychiatry and Neuroscience of Paris, INSERM U1266, Dynamics of Neuronal Structure in Health and Disease, Paris, France. [8]Université Paris Cité, Institute of Psychiatry and Neuroscience of Paris, INSERM U1266, NeurImag Core Facility, Paris, France. [9]Université Paris Cité, INSERM UMR-S1151, CNRS UMR-S8253, Institut Necker Enfants Malades, Paris, France. [10]Institut Curie, PSL University, Sorbonne Université, CNRS UMR144, Cell Biology and Cancer, Structure and Membrane Compartments, Paris, France. [11]Institut Curie, PSL University, Sorbonne Université, CNRS UMR144, Cell Biology and Cancer, Cell and Tissue Imaging Facility (PICT-IBiSA), Paris, France. [12]Penn State University College of Medicine, Cell and Biological Systems, Hershey, PA, USA. [13]GHU Paris Psychiatrie & Neurosciences, Paris, France. [14]These authors contributed equally: Pedro Dionisio, Quentin Lemercier. ✉e-mail: nuno.raimundo@psu.edu; thierry.galli@inserm.fr

