## [Transparent Peer Review file · Nature Communications]

VAMP7-dependent late endosomal secretion of ER and mitochondrial proteins impacts the tumor microenvironment and macrophage engagement

Corresponding Author: Dr Thierry Galli

Version 1:

Reviewer comments:

Reviewer #1

(Remarks to the Author)

Summary

This manuscript investigates a novel pathway of unconventional protein secretion (UPS) mediated by the SNARE protein VAMP7, focusing on the transport of tubular ER and mitochondrial components to late endosomes and their release. Using CRISPR/Cas9 knockout models in both epithelial and glioma cells, the authors demonstrate that VAMP7-dependent secretion is enhanced during ER and mitochondrial stress, functions independently of canonical degradative autophagy, and may act as an organelle quality control mechanism. In a glioma model, the loss of VAMP7 leads to increased tumor growth, suggesting a physiological relevance of this secretory pathway in cancer.

Major Comments:

Key conceptual questions:

The authors provided a graphical summary of this study in Figure 6I. However, the concept in this figure requires further verification and clarification.

- 1) Generally, the v-SNARE VAMP7 is located in the departing vesicles, while t-SNARES such as STX5, SNAP47, and STX17 are located in the target membranes and play a role in vesicle fusion at the target membrane. However, in Fig. 6I, their roles as classical SNARE molecules are unclear. Additionally, their orientation appears reversed; for instance, t-SNARES are shown in the vesicle-departing membranes rather than in the destination. How do the authors reconcile this finding, and what would be the roles of these SNARE molecules in this context?
- 2) What is the role of RTN3A in this process? Does it contribute to secretion, or is it simply a bystander cargo? In this and previous publications, the authors suggest that impaired autophagy by ATG5 KO can increase unconventional protein secretion. If RTN3 is indeed required for secretion, given that ATG5 is critical for autophagosome formation and that RTN3 bridges ER membranes with LC3-II-associated autophagosomal membranes, ATG5 KO would presumably inhibit RTN3-associated secretory autophagy.

Figure 1:

- 1) Panels E-I. How do the authors define the periphery and perinuclear regions? Additionally, in their quantifications, the authors employed Sholl analysis, typically used to measure neuronal morphology. However, the method details are not provided. A more detailed description of how perinuclear and peripheral CD63 quantification was performed is needed.
- 2) Panels E-I. The authors describe the perinuclear accumulation of CD63 as indicative of degradative compartments and its peripheral localization as reflective of secretory late endosomes. While this is an intriguing observation, the interpretation would benefit from additional experimental support. Specifically, co-localization studies with known markers of degradative or secretory compartments would substantiate these conclusions.
- 3) Panel J-K. The authors state that VAMP7 knockout (KO) increases the presence of ER whorls, suggesting a link to ER stress and impaired RTN3-mediated tubular ER dynamics. However, since the localization of RTN3 within these whorls has not been directly demonstrated, it would be helpful for the authors to clarify whether this association has been previously reported or experimentally confirmed in the current study. Additionally, the ER whorl structures are not clearly discernible in the provided ER tracker images. The inclusion of CLEM or EM images is highly recommended.

Figure 2:

- 1) Panels A-D. To more clearly assess whether VAMP7 knockout does not impair degradative autophagy, autophagic flux analysis using lysosomal inhibitors like Bafilomycin A1 is recommended. This approach would distinguish between increased autophagosome formation and impaired degradation.
- 2) Panels E-G. The images of WT cells treated with Torin show a strong green signal. Are these representative images appropriate? Additionally, are the RFP-GFP-LC3 signals observed in the nucleus genuine signals?

Figure 3:

Here and in other figures (e.g., S1 and S2), the authors suggest that both ATG5 KO and VAMP7 KO induce ER stress and mitochondrial damage, although they differentially affect secretion. While the authors provide some potential explanations in the discussion related to Figure 3 (lines 213–243), they do not offer a convincing account of how both increases and decreases in protein secretion can be associated with ER stress and mitochondrial damage. A more thorough discussion of how ATG5 KO and VAMP7 KO similarly induce ER stress and mitochondrial damage, yet differentially influence secretory processes, would help clarify the overall findings of the study.

Figure 4:

Have the authors examined whether Bafilomycin A1 (BafA1) treatment affects the secretion levels of endogenous RTN3 and VDAC in WT, VAMP7 KO, and ATG5 KO cells? Inclusion of these control data would provide an overall understanding of how lysosomal inhibition influences secretion across different genetic backgrounds, potentially clarifying the relationship between degradative autophagy and unconventional secretion.

Figure 5:

- 1) Panel A. The description of the short isoform of RTN3A requires further verification, as its domain composition differs across publications. A map with precise amino acid numbers and LIR and RHD locations is recommended. Does it contain LIR motifs? If so, an experiment with the shortest form of RTN3 lacking LIRs would help elucidate the role of LIRs in the RTN3-mediated pathway. If the authors' RTN3A contains LIRs, why is RTN3L, which supposedly has six LIRs, not involved in this process?
- 2) Panel C. The authors report an interaction between RTN3 and CD63. Interestingly, the observed molecular weight of CD63 in the immunoprecipitation (IP) results is approximately 50 kDa. Since the ER form of CD63 typically migrates at ~25–30 kDa, it remains unclear whether the interacting form corresponds to a Golgi-bypass ER form. Characterization of the glycosylation status (e.g., digestion with glycosidases or proteomic analysis) is recommended. Additionally, the authors could perform the IP experiment under Brefeldin A treatment, which disrupts ER-to-Golgi transport. If the interaction persists under these conditions, it would provide stronger evidence that an ER-localized, Golgi-bypass form of CD63 is involved in the interaction with RTN3.
- 3) Panel C and G. Consider adding marks indicating these IP results involve RTN3A and VAMP7, respectively, for clarity.

Figure 6:

- 1) Panel A. It is recommended to determine whether RTN3L and VDAC secretion is also induced by Thapsigargin.
- 2) Panel C. Similarly, investigating whether RTN3 secretion is induced by Antimycin is recommended.

Figure 7:

- 1) Panel A. There appears to be a discrepancy in the figure labels. Currently, the VAMP7 low (blue) group shows longer survival, which contradicts the text. Additionally, clarification on how “high” and “low” VAMP7 expression groups were defined would be helpful.

Minor Comments:

This manuscript contains several typos, including:

- 1) Figure 4A: “BaA1” should be corrected to “BafA1.”
- 2) Line 210: “VAMP7KO does induce major autophagy alteration...” should likely read “VAMP7KO does not induce major autophagy alteration.”
- 3) Line 71: “a N-terminal” should read “an N-terminal.”

Reviewer #2

(Remarks to the Author)

This manuscript entitled “Role of autophagy-dependent late endosomal secretion of ER and mitochondrial components in cell fitness and communication” by Vats S et al. presents a large amount of data regarding unconventional protein secretion in VAMP7 KO cells. I have a couple of inquiries.

1. In line 98, there is mention about Atlastin 3. However, I cannot find data regarding Atlastin 3 in the Results.
2. ER stress response can be seen in Atg5 KO cells in Fig. 3G. In line 136, can whorls be seen in Atg5 KO cells? It is recommended to add References for whorls as a marker of ER stress.
3. Is it correct to conclude that “VAMP7 KO does induce major autophagy alteration” in line 210, while there were only limited changes of autophagy in Fig. 2?
4. On the basis of the authors' results and papers, in what cells is the UPS significant and important particularly in relation to VAMP7 (e.g. secretory cells, most cells, cancer cells, etc?)

5. It is not clearly stated whether there is a statistically significant difference between curves of Fig. 7A.
6. The results of Fig. 7F are not clearly stated in the text.
7. It is not clearly stated in the text what is the reason for differences of tumor volume between Fig. 7E and 7G.
8. In Fig. 7A, it is not clear why high VAMP7 tumor has increased aggressiveness. Is there any effect on the tumor microenvironment? Are high VAMP7 cells more resistant to ER stress? (see inquiry 9)
9. Was cell death or survival of VAMP7 or Atg5 KO cells before and after treatment with ER stressor or mitochondrial stressors studied? If not, it is recommended to study cell death of VAMP7 KO cells in vitro.
10. In line 421, reference no might be 52.
11. The title is 'autophagy-dependent late ---'. However, late endosomal secretion appears to be parallel to autophagy and autophagy-independent as shown in Fig. 6I.

Reviewer #3

(Remarks to the Author)

The manuscript by Vats et al interrogates the inter-organellar trafficking of proteins that may direct tubular ER and mitochondrial content to multivesicular late endosome (aka MVB) potentially leading to their unconventional secretion. In particular, the paper focuses on VAMP7, which is a vesicular SNARE protein implicated in MVB mediated secretory pathway, and has a role in degradative autophagy, lysosomal secretion, and fusion of mitochondrial vesicles (MDVs) with endosome and lysosome. VAMP7 is not essential for survival in mice, though non-lethal anomalies have been observed, and authors have previously studied the role of VAMP7 in secretory processes in neurons and now expanded this to NRK cells and rat glioblastoma cells.

Given the role of stress and autophagy in unconventional secretion the authors' main approach was to generate VAMP7-deficient and ATG5-deficient NRK cells (VAMP7-KO and ATG5-KO respectively) and to compare their secretory properties and impact on mitochondria, ER, as well as exchange of organellar markers between these structures and CD63+ MVBs. The study also examines the transcriptomes of these cell lines wild type and KO cells, and extracts predicted pathways affected by VAMP7 and ATG5 deficiency. Overall, this analysis relies mainly on microscopic colocalization studies, proximity ligation and co-immunoprecipitation of some of the implicated molecular players.

Among more notable observations is the evidence that the secretome of NRK cells contains elements of the ER, such as RTN3 isoform, and this is depleted in the case of VAMP7-KO cells, but enhanced in the case of ATG5-KO cells with deficient autophagy pathway. Similar pattern was detected also for mitochondrial proteins such as VDAC leading the authors to propose a role for these pathways in trafficking organellar content to the MVB. Indeed, VAMP7 is found to regulate the protein transport between ER and endosomes, where RTN3 interacts with CD63. This transport is upregulated in settings of mitochondrial and ER stress. Notably, while the cellular responses in the context of VAMP7 and ATG5 perturbations seem interrelated, no viable cells were obtained while genetically targeting both proteins at the same time. Both VAMP7 and ATG5 deficiencies impacted mitochondrial respiration. Finally, authors explore the consequences of VAMP7 and ATG5 targeting in the rat glioblastoma cell line RG2. Interestingly, but not unexpectedly, in this case ATG5 deficiency diminishes tumor growth in the brain, while VAMP7-KO tumors were more necrotic, which the authors attribute to the roles of these proteins in what they refer to as "pathological stress".

Overall, the study targets and interesting nexus between cellular stress, autophagy and the role of VAMP7 in inter-organellar trafficking. However, while the data are interesting, as presented, the paper suffers from several shortcomings.

1. The most obvious problem is that while nominally the study pertains to unconventional secretion, the nature of this secretion is only scarcely covered in Fig. 1. One may guess that finding ER and mitochondrial proteins in the cellular secretome is a function of exosomal release, but this is never formally or directly shown, characterized (MISEV2023) functionally linked to MVB, or explored for function.
2. Further to this point, the authors place several intriguing hints in the paper but many are not fully investigated. The case in point is the title of the manuscript: "Role of Autophagy-Dependent Late Endosomal Secretion of ER and Mitochondrial Components in Cell Fitness and Communication", which suggests studies on cellular fitness and communication neither is investigated in depth.
3. The content of ER (Bare et al 2025) and mitochondrial (Ancuzo et al 2021) proteins in extracellular vesicles (EVs) has already been proposed, and to some extent investigated, as have various points of contact and communication between organelles, including EV-generating, VAPA-dependent membrane contact sites (Barman et al 2022). While the current study adds to this body of work, this is mainly in a form of supporting detailed evidence.
4. The 'bifocal' nature of the present study tackling both VAMP7 and ATG5 is not ideal and makes the work look unfocused. Although these pathways are interlinked and relevant as documented by the absence of viable double knock-out cells, the nature and reason for this observation is not fully explained. These are also different proteins and pathways that are being compared, especially since their presented connections are often inferential. This, perhaps, overly broad scope results in somewhat descriptive nature of the analysis and the abundance of hypotheses and speculations.
5. A related point is that while the authors invoke various cellular and pathological "stresses" the study does not have a defined biological direction where these stresses would be relevant for larger functions. For example, the impact of changes in question on specific cellular or tissue phenotypes, such as survival, metabolism, function, are not really explored to any

greater depth. The present study, as presented, would unlikely explain the pathological alterations in VAMP7-deficient mice.

6. This point is also reflected in the study on glioblastoma, which is woefully inadequate and of uncertain validity. While the observations of altered tumor growth, necrosis and progression are intriguing they are not mechanistically explained. There is no direct evidence as to the role of cell-autonomous or secretory events in these changes, and neither is there any convincing documentation that VAMP7 is a major force in the context of highly heterogeneous landscape of human glioblastoma (Wen et al 2020), let alone “tumor microenvironment”. Although the authors include Kaplan Meier curves documenting some difference in survival of glioblastoma patients with VAMP7-high and VAMP7– low tumor characteristics, this may represent the effect of broader molecular subgroups rather than VAMP7 itself. Why would patients with low VAMP7 expression have a survival advantage? Linking this to experiments with a single rodent glioma cell line is probably an overextension.

7. The authors make frequent and consequential references to their previous work which the reader is not obligated to follow. The paper is also written in a somewhat cryptic and meandering manner, such that it is often difficult to extract the line of authors' reasoning, a sense that is amplified by suboptimal expressions and sometimes overly bold statements. For example:

- Page 5: Experiments with NRK cells are hardly a basis “generalization” of VAMP7 effects
- Page 5: Expression: “localization of CD63 to the perinuclear riation found in WT cells” – should this be “perinuclear compartment”?
- Page 7: “These results showed that VAMP7KO does induce major autophagy alteration” – should this be: “does not”?
- Page 10: Exosomal CD63 often appears as smear in many studies, so the speculation as to its role is unfounded.
- Page 12: The statement: “We found a tendency towards larger tumors for VAMP7KO RG2 cells which was not visible in fixed tumors, likely due to removal of tumor interstitial fluid by fixation (Fig 7D, E)” exemplifies ambivalent and speculative comments that add little to the content.

Reviewer #4

(Remarks to the Author)

Version 2:

Reviewer comments:

Reviewer #1

(Remarks to the Author)

The authors have adequately addressed most of my initial concerns.

One remaining point I would like to note relates to the discussion of RTN3A and RTN3L. In lines 501–510, the revised text suggests that RTN3A is primarily associated with protein secretion, whereas RTN3L is involved in protein degradation. However, a recent study suggests that RTN3L may also play a role in protein secretion (PMID: 39919755). Therefore, it is recommended that this discussion be either removed or rewritten to better reflect the current understanding.

Reviewer #2

(Remarks to the Author)

This revised manuscript entitled “Role of VAMP7-dependent late endosomal secretion in the control of endoplasmic reticulum and mitochondrial homeostasis” by Vats S et al. has been improved by incorporation of the reviewer’s inquiries. I still have a couple of inquiries.

1. In the answer to the inquiry 2 of Reviewer 2, the reference number could be 38 instead of 20.
2. In line 441-448, the most important result is the increase of tumor volume of VAMP7 KO cell graft. Then, other findings were described earlier and in more detail.
3. In Fig. 7M, it is not clear what the y axis label (%Macrophages in tumor) represents.
4. In Fig. 7N, there was no significant difference of %macrophage in tumor periphery between WT and VAMP7 KO. However, in line 445-6, ‘macrophage restriction to the periphery of VAMP7 KO’ was mentioned.
5. Even if macrophage was restricted to the periphery, it could be the consequence of tumor necrosis in the central area of the tumor.
6. Even if tumor size was different between WT & VAMP7, there is no evidence that the difference is due to changes in innate immunity in response to altered release of stress-related molecules.
7. In lines 552-555, there is no evidence that increased necrotic area is due to reduced macrophage-mediated removal of debris.
8. In Fig. 1H-I, it is not clearly seen whether the distance from the nucleus is increased in VAMP7 KO cells. Probably, broken circular line in Fig. 7H is plasma membrane boundary. It may be helpful to circle nuclei.

Reviewer #3

(Remarks to the Author)

The revisions and rebuttals following the prior review of the manuscript by Vats et al address large proportion of initial concerns. It is a nice paper. While one may agree with much of the authors' argumentation as to inter-organellar interactions, and secretory effects of VAMP7 the glioblastoma (GBM) aspect of the study continues to be a stretch for several reasons:

1. A single rodent cell line is a far cry from the enormous heterogeneity of human GBM, for which proper models would include PDX, or human glioma stem cell isolates, and definitely more than one to make any meaningful conclusions. What subtype of brain tumours does the RG2 cell line represent (transcriptome, markers, lineage identity, properties)?
2. The nomenclature of brain tumours used by the authors is extremely confusing. The current (2021) WHO classification moves away from TCGA subtypes, which include considerable proportion of IDH1 mutant tumours, mainly in the proneural subtype. IDH1 mutant tumours are no longer included in the GBM diagnosis and possess vastly different clinical and cellular characteristics than GBM (Wen et al 2020). This is never clarified in the paper, which in Supplemental Figure 7 contains a random mix of low and high grade tumours from different cohorts and of different origins, with names such as "diffuse astro glioma" (?), some of which are simply not in use. In addition, in Fig. 7 along with the corresponding text and rebuttal the authors single out TCGA Stupp tumours. What are they? Stupp protocol is a standard of care for virtually all newly diagnosed IDH-WT GBMs. Are the data representative of recurrent tumours? Is this the reason why VAMP7 no longer separates survival patterns? Which cells in the tumours express VAMP7? Can this be inferred from single cell datasets which are readily accessible? Are the authors suggesting that VAMP7 is uniquely linked to brain tumours or is this a pathway of many cancer cells under metabolic stress? The GBM aspect of the present study should be either qualified, as preliminary (which it is) or strengthened and properly presented.

Reviewer #4

(Remarks to the Author)

Version 3:

Reviewer comments:

Reviewer #3

(Remarks to the Author)

The recent revisions and rebuttals of the manuscript nominally address the prior questions, and at this stage of these revision, the proposed changes go as far as it is practical to alleviate the reviewer's concerns. Whether VAMP7 is truly a meaningful element in progression of GBM and what the different comparisons presented in the paper really mean is an open question, but this reviewer appreciates the effort on the part of the authors.

We are grateful to the reviewers for their insightful and constructive comments. In response, we have performed additional experiments, reanalysed existing datasets, and thoroughly revised all sections of the manuscript. Our detailed responses are provided below. The main textual changes and additions are highlighted in the revised manuscript.

Reviewer #1 (Remarks to the Author):

Summary

This manuscript investigates a novel pathway of unconventional protein secretion (UPS) mediated by the SNARE protein VAMP7, focusing on the transport of tubular ER and mitochondrial components to late endosomes and their release. Using CRISPR/Cas9 knockout models in both epithelial and glioma cells, the authors demonstrate that VAMP7-dependent secretion is enhanced during ER and mitochondrial stress, functions independently of canonical degradative autophagy, and may act as an organelle quality control mechanism. In a glioma model, the loss of VAMP7 leads to increased tumor growth, suggesting a physiological relevance of this secretory pathway in cancer.

We are grateful for these complimentary comments.

Major Comments:

Key conceptual questions:

The authors provided a graphical summary of this study in Figure 6I. However, the concept in this figure requires further verification and clarification.

1) Generally, the v-SNARE VAMP7 is located in the departing vesicles, while t-SNAREs such as STX5, SNAP47, and STX17 are located in the target membranes and play a role in vesicle fusion at the target membrane. However, in Fig. 6I, their roles as classical SNARE molecules are unclear. Additionally, their orientation appears reversed; for instance, t-SNAREs are shown in the vesicle-departing membranes rather than in the destination. How do the authors reconcile this finding, and what would be the roles of these SNARE molecules in this context?

The concept of vesicle (v-SNAREs) and target (t-SNAREs) was conceived for exocytosis, i.e., the fusion of secretory vesicles with the plasma membrane. It does not apply as strictly to homotypic endosome fusion or to intracellular anterograde and retrograde transport pathways. The SNAREs have been structurally reclassified based on their contributions to the ionic zero layer. R-SNAREs provide an arginine (R) to this ionic layer. Q-snares provide complementary glutamines (Q) ¹ and hence their orientation in this regard is not of a significant consequence. Here, the role proposed for VAMP7, which resides on late endosomes, is to mediate fusion with ER-derived vesicles bearing STX5, SNAP47, in the same way that Fon and colleagues found that mitochondria-derived vesicles (MDVs) fuse with late endosomes using the VAMP7-STX17-SNAP29 SNARE complex, STX17-SNAP29 residing in the membrane of MDVs ². We have clarified this point in the legend of our graphical representation (Supp. Fig. 6E) and the text.

2) What is the role of RTN3A in this process? Does it contribute to secretion, or is it simply a bystander cargo? In this and previous publications, the authors suggest that impaired autophagy by ATG5 KO can increase unconventional protein secretion. If RTN3 is indeed required for secretion, given that ATG5 is critical for autophagosome formation and that RTN3 bridges ER membranes with LC3-II-associated autophagosomal membranes, ATG5 KO would presumably inhibit RTN3-associated secretory autophagy.

The short form of RTN3, RTN3A, might play a role in VAMP7-dependent late endosomal secretion; however, in the current manuscript, we project it as a bystander cargo. Of note,

we previously found that other reticulons and atlastins are secreted via a mechanism similar to that of RTN3A.

Here, we demonstrate that the short form of RTN3, RTN3A, is secreted by epithelial and glioma cells. In contrast, the long form, RTN3L, is not secreted but plays a role in degradative autophagy³. Thus, the mechanism of VAMP7-dependent late endosomal secretion we propose is distinct from secretory autophagy, i.e., the secretion of autophagosomes, which is mediated by Sec22b⁴. RTN3A has 2 LIR motifs, whereas RTN3L has 6 LIR motifs. Increased secretion of RTN3A, mediated by VAMP7 (Fig. 1B-E, Fig. 4A,B) is likely the signature of a compensation for the lack of degradative reticulophagy in ATG5KO. Therefore, the two isoforms of RTN3 appear as a good illustration of these parallel mechanisms of secretion (RTN3A) and degradation (RTN3L). The potential role of the long amino-terminal tail of RTN3L in distinguishing secretion from degradation will certainly merit further attention. We have further emphasized this critical point in the text. Please see page 16.

Figure 1:

1) Panels E-I. How do the authors define the periphery and perinuclear regions? Additionally, in their quantifications, the authors employed Sholl analysis, typically used to measure neuronal morphology. However, the method details are not provided. A more detailed description of how perinuclear and peripheral CD63 quantification was performed is needed. 2) Panels E-I. The authors describe the perinuclear accumulation of CD63 as indicative of degradative compartments and its peripheral localization as reflective of secretory late endosomes. While this is an intriguing observation, the interpretation would benefit from additional experimental support. Specifically, co-localization studies with known markers of degradative or secretory compartments would substantiate these conclusions.

We thank the reviewer for this comment. We have now employed a more detailed method to quantify the perinuclear/peripheral localisation of CD63. We have utilised an ImageJ plugin for the same (Fig.1H, I). The plugin details and the way we defined the perinuclear/peripheral area have now been added to the methods section of the manuscript. Using this home-designed plugin, we were able to demonstrate more convincingly the strong effect of VAMP7KO on CD63 intracellular distribution, which is reversed by re-expression of VAMP7 (Fig. 1H, I).

To analyse the effect of VAMP7 on lysosomes, we performed a functional assay of lysosomal pH. pH levels in lysosomes vary depending on their position within the cell, which can serve as a proxy for their degradative capacity^{5,6}. As expected, peripheral lysosomes closer to the plasma membrane had higher pH, indicating reduced degradative potential, independent of cell genotype (Fig. 1J, K). Mounting evidence supports distinct roles for lysosomes depending on their intracellular position, with perinuclear lysosomes widely regarded as more degradative and having a lower pH. In contrast, peripheral ones may be more linked to other functions, including secretion^{7,8,9}. These concordant experimental and literature-based observations may thus suggest that recruitment of CD63+ vesicles from the perinuclear region to the periphery in VAMP7KO NRK cells has functional consequences.

3) Panel J-K. The authors state that VAMP7 knockout (KO) increases the presence of ER whorls, suggesting a link to ER stress and impaired RTN3-mediated tubular ER dynamics. However, since the localization of RTN3 within these whorls has not been directly demonstrated, it would be helpful for the authors to clarify whether this association has been previously reported or experimentally confirmed in the current study. Additionally, the ER whorl structures are not clearly discernible in the provided ER tracker images. The inclusion of CLEM or EM images is highly recommended.

We thank the reviewer for this request. The VAMP7KO cells have increased levels of ER stress (Fig. 2C-I) and an increased presence of ER whorls (Fig. 2 F-I). We have not studied the localisation of RTN3 in these whorls because immune-EM studies would considerably delay publication without being so informative at this point. However, we have reanalysed the EM in depth in the NRK cells. We found that the EM of the VAMP7KO cells shows an accumulation of ER whorls (Fig. 2H, I). Together with transcriptomics and protein expression, we are now confident in stating that VAMP7 KO induces ER stress (Fig. 2).

Figure 2:

1) Panels A-D. To more clearly assess whether VAMP7 knockout does not impair degradative autophagy, autophagic flux analysis using lysosomal inhibitors like Bafilomycin A1 is recommended. This approach would distinguish between increased autophagosome formation and impaired degradation.

We have added autophagy assays with Bafilomycin A1 in the revised manuscript (Supp. Fig. 2). These experiments show that VAMP7 does not play a major role in degradative autophagy as expected.

2) Panels E-G. The images of WT cells treated with Torin show a strong green signal. Are these representative images appropriate? Additionally, are the RFP-GFP-LC3 signals observed in the nucleus genuine signals?

We have changed the representative image for the autophagy RFP-GFP-LC3 assay (Supp. Fig. 2G); however, the nuclear accumulation of LC3 is well documented and has been reported previously¹⁰. The latter article showed that “an acetylation-deacetylation cycle ensures that LC3 effectively redistributes in an activated form from nucleus to cytoplasm, where it plays a central role in autophagy to enable the cell to cope with the lack of external nutrients”.

Figure 3:

Here and in other figures (e.g., S1 and S2), the authors suggest that both ATG5 KO and VAMP7 KO induce ER stress and mitochondrial damage, although they differentially affect secretion. While the authors provide some potential explanations in the discussion related to Figure 3 (lines 213–243), they do not offer a convincing account of how both increases and decreases in protein secretion can be associated with ER stress and mitochondrial damage. A more thorough discussion of how ATG5 KO and VAMP7 KO similarly induce ER stress and mitochondrial damage, yet differentially influence secretory processes, would help clarify the overall findings of the study.

As we explained in the discussion of our manuscript (previously lines 430-455), VAMP7 regulates late endosomal secretion of ER and mitochondrial proteins, whereas ATG5 is an essential autophagy protein that participates in the autophagy-mediated degradation of surplus or damaged ER and mitochondrial proteins. These two pathways allow cells to respond to ER and mitochondrial stress, and, as shown in our manuscript, the increase in VAMP7-dependent secretion is adaptive. It is elevated when degradative autophagy is absent, as in ATG5KO cells or upon Bafilomycin A1 treatment (Fig. 1B-E, Fig. 6A-H). Hence, even though these two proteins have an opposite effect on secretion, the overall impact on the decrease of ER and mitochondrial fitness in the absence of either can be predicted reasonably well.

Figure 4:

Have the authors examined whether Bafilomycin A1 (BafA1) treatment affects the secretion levels of endogenous RTN3 and VDAC in WT, VAMP7 KO, and ATG5 KO cells? Inclusion of these control data would provide an overall understanding of how lysosomal inhibition influences secretion across different genetic backgrounds, potentially clarifying the

relationship between degradative autophagy and unconventional secretion.

We fully agree with the reviewer. Here, we indeed have examined the effect of Bafilomycin A1 (BafA1) treatment on the secretion levels of endogenous RTN3 and VDAC in WT, VAMP7KO, and ATG5KO cells (Fig. 1B-E). We apologise for the lack of clarity on this point in the text. We have rectified this issue in the revised manuscript.

Figure 5:

1) Panel A. The description of the short isoform of RTN3A requires further verification, as its domain composition differs across publications. A map with precise amino acid numbers and LIR and RHD locations is recommended. Does it contain LIR motifs? If so, an experiment with the shortest form of RTN3 lacking LIRs would help elucidate the role of LIRs in the RTN3-mediated pathway. If the authors' RTN3A contains LIRs, why is RTN3L, which supposedly has six LIRs, not involved in this process?

As explained above, the short form of RTN3, RTN3A, might play a role in VAMP7-dependent late endosomal secretion; however, in the current manuscript, we project it as a bystander cargo. The short form of RTN3, RTN3A, is secreted, whereas the long form RTN3L is degraded by degradative autophagy^{3,11}. The mechanism of VAMP7-dependent late endosomal secretion that we propose is not identical to secretory autophagy. In our manuscript, we suggest that RTN3A is delivered to late endosomes via ER-derived vesicles; therefore, the LC3-binding capacity of RTN3, as well as the presence/absence of LIR motifs, is not the key determinant. We think the extended amino terminus of RTN3L might be critical for directing ER tubules toward degradative autophagy. This raises an essential question beyond the scope of the present study, certainly for future studies. We have included a sentence in the discussion to emphasize this point (page 16).

2) Panel C. The authors report an interaction between RTN3 and CD63. Interestingly, the observed molecular weight of CD63 in the immunoprecipitation (IP) results is approximately 50 kDa. Since the ER form of CD63 typically migrates at ~25–30 kDa, it remains unclear whether the interacting form corresponds to a Golgi-bypass ER form. Characterization of the glycosylation status (e.g., digestion with glycosidases or proteomic analysis) is recommended. Additionally, the authors could perform the IP experiment under Brefeldin A treatment, which disrupts ER-to-Golgi transport. If the interaction persists under these conditions, it would provide stronger evidence that an ER-localized, Golgi-bypass form of CD63 is involved in the interaction with RTN3.

We thank the reviewer for this insightful suggestion, which raises, in fact, a complex issue.

Indeed, newly synthesized CD63 was shown to traffic from the ER to the Golgi to the cell surface, then to be endocytosed by AP2/clathrin-mediated endocytosis, transferred to endosomes, and then to late endosomes by a mechanism involving AP3¹². The interaction we have found between RTN3A and CD63 does not necessarily imply that the proteins travel together from the ER to late endosomes, suggesting that CD63 bypasses the Golgi. The topologies of CD63 (4 transmembrane domains, Figure A) and RTN3A (4 membrane spanning regions, Figure B) are such that RTN3A cannot interact with the glycosylated part of CD63, only with the transmembrane domains and/or the cytosolic domains of CD63.

Figure A. Schematic representation of the probable structure of CD63. In orange, the cysteine residues that form the three disulfide bridges are depicted. In blue, the N-glycosylation sites present in the large extracellular loop of CD63 are represented ¹³

Figure B. Schematic representation of the structure of RTN3 ¹⁴

These principles being set, we still decided to further explore this critical question with new experiments. In our original manuscript, we had performed an immunoprecipitation of the Myc-tagged short form of RTN3, Myc-RTN3A, and showed an interaction with endogenous CD63. We used a well-characterised and published AD1 monoclonal mouse anti-rat CD63 antibody in our studies; however, in rat cells, AD1 recognises only the native glycosylated form of CD63 ¹⁵, as we now confirm.

Indeed, the AD1 antibody does not recognize endogenous CD63 if the sample has been

denatured in SDS sample buffer by boiling or deglycosylated (Supp. Fig. 6B).

To confirm that the 50 kDa band corresponds to the glycosylated form of CD63, we treated the samples with deglycosylating enzymes, Endoglycosidase H (Endo H) and PNGase F, obtained from New England Biolabs (NEB). PNGase F is expected to fully deglycosylate CD63 by removing all types of N-linked (Asparagin-linked) glycosylation, i.e., high mannose, hybrid, bi-, tri-, and tetra-antennary (Figure D), depending on the length of the treatment. Here, we used Rapid PNGase F, which is supposed to enable complete deglycosylation in 10 minutes following the manufacturer's protocol. Endo H is a highly specific endoglycosidase that cleaves only high mannose and some hybrid types of N-linked oligosaccharides, but not highly processed complex oligosaccharides from glycoproteins passing by the Golgi apparatus (Figure C-D). Upon entering the ER, a molecule containing 14 sugar subunits is linked en bloc to an asparagine in a selective manner by the enzyme oligosaccharyl transferase. It is this oligosaccharide molecule, which is modified by a series of enzymes as the protein moves through the different compartments of the Golgi apparatus. Endo H can cleave each structure of this oligosaccharide as it is processed until the enzyme Golgi α -mannosidase II removes two mannose subunits. Since all later oligosaccharide structures are resistant to cleavage by Endo H, the enzyme is widely used to report the extent of oligosaccharide processing a protein of interest has undergone. Thus, Endo H

sensitivity has been associated with Golgi bypass. Samples were treated with Endo H at 37°C for 90 minutes following NEB's protocol.

Figure C. N-linked glycan processing in the endoplasmic reticulum and Golgi apparatus and Endo H/PNGase activities. Adapted from ¹⁶ N-glycans that are Endo H-sensitive are shown in the red box, including high-mannose and hybrid glycans that have two terminal mannose residues, which is required for recognition by Endo H and its activity (red oval). N-glycans that are PNGase-sensitive are shown in the blue box, i.e. all N-glycans. Asn, asparagine; ER, endoplasmic reticulum; Fuc, fucose; Gal, galactose; Glc, glucose; GlcNAc, N-acetylglucosamine; Man, mannose; Sia, sialic acid.

Figure D. Scheme of PNGase F and Endo H digestion ¹⁷ PNGase F cuts the bond between GlcNAc and Asn, liberating the entire sugar chain and converting Asn into Asp. At the contrary, Endo H cleaves the link between the two GlcNAc residues in the core region, leaving one GlcNAc still bound to the protein. X and Y are unspecified sugar residues.

Unfortunately, a key step in the Endo H treatment is boiling the samples before treatment with the enzyme, which failed due to the inability of AD1 to bind to denatured CD63 (Supp. Fig. 6B). Therefore, for the revision and to answer the question raised by the reviewer successfully, we chose to proceed with a Co-IP of GFP-CD63 and Myc-RTN3A. The samples were deglycosylated following to immunoprecipitation to demonstrate that RTN3A interacted with a glycosylated form of CD63. Here we found that RTN3A pulled down both unglycosylated CD63 and glycosylated forms of the protein (Fig 4E). However, this might be

due to overexpression of the GFP-tagged form, and in normal conditions, most, if not all, CD63 might not bypass the Golgi.

Additionally, we performed the IP experiment under Brefeldin A (BFA) treatment, which disrupts ER-to-Golgi transport. Of note, it was previously shown that “newly-synthesised CD63 was susceptible to digestion with Endo H, and the protein was not completely resistant to endoglycosidase H until after 4 hours of chase, indicating that transport through the medial and trans-Golgi complex with conversion of high-mannose carbohydrates to complex oligosaccharide side chains had occurred. This finding suggests a relatively long processing time for CD63¹⁸. Long treatment with BFA, however, induces the collapse of all intracellular membranes, including endosomes¹⁹. Therefore, BFA sensitivity is not necessarily a marker of Golgi-bypass in the case of CD63.

Here, based on additional experiments, we found that the interaction of RTN3A and GFP-CD63 persists following a 90-minute treatment with BFA at 20 µg/ml (Fig 4E). As the reviewer mentioned, this might provide clear evidence that an ER-localized, Golgi-bypass form of CD63 can interact with RTN3. Again, this might be due to overexpression of the GFP-tagged form, and in normal conditions, most, if not all, CD63 might not bypass the Golgi. We have included these new experiments and discussed this issue in greater detail. Our conclusion remains that RTN3A and CD63 can interact, that this interaction depends on VAMP7-dependent transport (Fig. 4C-E), and it is a mark of RTN3A reaching late endosomes. Whether the pool of CD63 that interacts with RTN3A has bypassed the Golgi remains an open question due to limitations in currently available approaches. We have discussed the new results in the light of the abovementioned limitations.

3) Panel C and G. Consider adding marks indicating these IP results involve RTN3A and VAMP7, respectively, for clarity.

We have done the needful in the revised manuscript.

Figure 6:

- 1) Panel A. It is recommended to determine whether RTN3 and VDAC secretion is also induced by Thapsigargin.
- 2) Panel C. Similarly, investigating whether RTN3 secretion is induced by Antimycin is recommended.

We thank the reviewer for this insightful suggestion. We have now added data on how RTN3 and VDAC secretion is impacted by Antimycin A and Thapsigargin, and *vice versa* (Fig. 6A-H). We found that Antimycin A only stimulated secretion of VDAC, whereas Thapsigargin only stimulated that of RTN3A, thus suggesting that specific stress induced specific transport to late endosomes and further secretion.

Figure 7:

- 1) Panel A. There appears to be a discrepancy in the figure labels. Currently, the VAMP7 low (blue) group shows longer survival, which contradicts the text. Additionally, clarification on how “high” and “low” VAMP7 expression groups were defined would be helpful.

We apologise for the error in the Kaplan-Meier curve of Fig. 7A, the low VAMP7 group has more aggressiveness and a lower survival. In the original version of this article, this panel was unfortunately mislabelled at a late stage, and we missed this error.

This question prompted the reviewer to further analyse our data and publicly available data in greater detail. We found that in the GBM-Mark cohort, high expression levels of VAMP7 were associated with better survival (Fig. 7A). However, to a lesser extent, in the TCGA-GBM database, on the subset of tumors exposed to Stupp treatment (Fig. 7B). Lastly, the

TCGA-GBMLGG database tumor subtypes were also evaluated. Although global expression of VAMP7 was similar in mesenchymal or classical GB tumors, there was a significantly extended survival associated with high VAMP7 expression in the mesenchymal subtype (which was not the case for classical tumors, Fig. 7C). In addition, we have also tested the association of VAMP7 expression and patients' survival for other primary brain tumors. This indicated that, across many independent cohorts and primary brain tumor types, higher VAMP7 expression was associated with better survival. This data was added in the supplemental material in the revised version of the manuscript (Fig. 7B-F). We also analysed patients' survival data for RTN3. In both the TCGA-GBMLGG dataset and the CGGA dataset, we observed that high expression levels of RTN3 were associated with better patient' survival. We have now added these new results in this revision in the supplemental material (Suppl. Fig. 7G and 7H) as they further support our conclusions.

Minor Comments:

This manuscript contains several typos, including:

- 1) Figure 4A: "BaA1" should be corrected to "BafA1."
- 2) Line 210: "VAMP7KO does induce major autophagy alteration..." should likely read "VAMP7KO does not induce major autophagy alteration."
- 3) Line 71: "a N-terminal" should read "an N-terminal."

We thank the reviewer for carefully studying our manuscript. We have fixed these errors in the revised version of the manuscript.

Reviewer #2 (Remarks to the Author):

This manuscript entitled “Role of autophagy-dependent late endosomal secretion of ER and mitochondrial components in cell fitness and communication” by Vats S et al. presents a large amount of data regarding unconventional protein secretion in VAMP7 KO cells. I have a couple of inquiries.

We thank the reviewer for their efforts in reviewing our manuscript and for providing their critical feedback.

1. In line 98, there is mention about Atlastin 3. However, I cannot find data regarding Atlastin in the Results.

We were referring to our previously published work ¹¹. We have removed Atlastin from the current manuscript as it was not studied here.

2. ER stress response can be seen in Atg5 KO cells in Fig. 3G. In line 136, can whorls be seen in Atg5 KO cells? It is recommended to add References for whorls as a marker of ER stress.

Indeed, ER whorls were observed in ATG5 KO cells and quantified (Fig. 2E, I) based on our imaging data (Fig. 2F, H). However, both conventional and electron microscopy analyses failed to reveal any significant differences in ER morphology between ATG5 KO and control cells.

We apologise for missing this reference, we have added it in the revised manuscript ²⁰.

3. Is it correct to conclude that “VAMP7 KO does induce major autophagy alteration” in line 210, while there were only limited changes of autophagy in Fig. 2?

We apologise for the error in the text; we meant to conclude that VAMP7KO does not induce a significant alteration in autophagy. We have made the necessary edits to rectify this error.

4. On the basis of the authors’ results and papers, in what cells is the UPS significant and important particularly in relation to VAMP7 (e.g. secretory cells, most cells, cancer cells, etc?)

Building on the current manuscript and our previous work, we suggest that VAMP7-dependent unconventional protein secretion (UPS) is a broadly conserved mechanism operating in diverse cell types (e.g., PC12, NRK, RG2), especially when autophagic flux is reduced. This study provides, for the first time, evidence supporting a possible in vivo role for this pathway in cancer, where tumor cells represent a paradigm of chronic cellular stress. We have further emphasized this critical finding in the discussion.

5. It is not clearly stated whether there is a statistically significant difference between curves of Fig. 7A.

We apologize for this omission and have added the statistics in the graph (Fig. 7A).

6. The results of Fig. 7F are not clearly stated in the text.

7. It is not clearly stated in the text what is the reason for differences of tumor volume between Fig. 7E and 7G.

8. In Fig. 7A, it is not clear why high VAMP7 tumor has increased aggressiveness. Is there any effect on the tumor microenvironment? Are high VAMP7 cells more resistant to ER stress? (see inquiry 9)

Please allow us to address these three points together.

We apologise for the error in the Kaplan-Meier curve of Fig 7A., the low VAMP7 group has more aggressiveness and a lower survival. We have rectified this in the current manuscript.

We have addressed this point by performing new experiments to measure macrophage and neutrophil invasion, and we have rewritten the discussion to emphasize the results related to VAMP7KO.

Here, we observed smaller tumors in mice implanted with ATG5KO RG2 cells than in those implanted with WT or VAMP7 KO cells. This is consistent with previous studies showing reduced growth of autophagy-deficient tumors^{21,22}. The ATG5 KO phenotype likely reflects the limited metabolic flexibility and stress resistance of cancer cells in the absence of autophagy, which normally supports tumor growth under nutrient deprivation or hypoxia²³. We also found fewer macrophages in ATG5KO tumors, though the effect was less pronounced than in VAMP7KO tumors. This suggests that autophagy contributes to macrophage recruitment, but that VAMP7-dependent pathways exert a broader influence on tumor-immune communication, encompassing both autophagy-dependent and unconventional secretion mechanisms.

Interestingly, VAMP7KO tumors, although larger, exhibited increased necrosis, which may result from reduced macrophage-mediated debris removal, defective stress-adaptive secretion, and impaired lipid clearance²⁴. Consistent with this, macrophages were restricted to the periphery of VAMP7 KO tumors, whereas neutrophil infiltration was unaffected (Fig. 7L-N, Supp. Fig. 7I-J). Our findings support the notion that late endosomal secretion is a critical determinant of macrophage recruitment to tumors. The attenuation of ER- and mitochondria-derived protein secretion in VAMP7KO tumors likely reduces the release of stress-associated signals that may contribute to macrophage attraction to aggressive tumor regions. We propose a mechanism through which macrophages could detect cellular stress, a mechanism of high relevance in maintaining tissue homeostasis²⁴. Since GB-associated macrophages predominantly exhibit immunosuppressive functions, it is reasonable to propose that altering VAMP7-dependent secretion could reshape the immune landscape of inherently “cold” tumors such as GB. Such reprogramming may, in turn, create a therapeutic window that enhances the efficacy of immune checkpoint inhibitors. It will be particularly relevant to test if VAMP7 mediates the presentation of mitochondria and ER peptides by MHC class I. We did not observe an opposite phenotype in ATG5KO tumors associated with macrophage invasion (Fig. 7L-N), but this might be due to decreased tumor size (Fig. 7F-I) or to differences in macrophage subtypes. Indeed, VAMP7KO tumors may limit infiltration by tumor-suppressive macrophages while retaining tumor-supportive macrophages at the periphery. Further studies will be required to clarify these differential effects and the underlying mechanisms.

9. Was cell death or survival of VAMP7 or Atg5 KO cells before and after treatment with ER stressor or mitochondrial stressors studied? If not, it is recommended to study cell death of VAMP7 KO cells in vitro.

We have performed cell survival assays at varying doses of Antimycin A and Thapsigargin (Figure E). We observe that the dose that we chose in our studies does not have any toxic effects on the cells (Figure E). At much higher dose, we can see a differential toxicity for Antimycin A versus Thapsigargin. We have not added this data in the paper figures; however, if the reviewer feels that we need to add this in the manuscript, we will do so.

Figure E. Cytotoxicity of Antimycin A and Thapsigargin on WT, ATG5KO and VAMP7KO NRK cells with LDH-Glo Cytotoxicity Assay from Promega. Top, raw data for recorded Luminescence. Bot, cytotoxicity calculated using this formula:

$$100 \times \frac{(\text{Experimental LDH Release} - \text{Medium Background})}{(\text{Maximum LDH Release Control} - \text{Medium Background})}$$

“Lactate Dehydrogenase (LDH) catalyzes the oxidation of lactate with concomitant reduction of NAD⁺ to NADH. Reductase uses NADH and reductase substrate to generate luciferin, which is converted to a bioluminescent signal by Ultra-Glo™ rLuciferase. The light signal generated is proportional to the amount of LDH present, released into the cell culture medium upon disruption of the plasma membrane of dead cells”. In brief, following the manufacturer’s protocol, 5000 NRK cells in 100 μl of culture medium containing serum were treated with increasing concentrations of Antimycin A or Thapsigargin. Samples (3μl) were collected for each well at 24 hours and diluted at 1:100 in LDH Storage Buffer (200mM Tris-HCl (pH 7.3), 10% Glycerol, 1% BSA). 50 μl of the diluted samples were combined with LDH Detection Reagent. Luminescence was recorded after 60 minutes incubation at room temperature. Two-way ANOVA with post-hoc Tukey’s multiple comparisons test (compare row means, main row effect). ****, p<0.0001. Arrows indicate Antimycin A/Thapsigargin concentration used for secretion experiment in Fig. 6.

10. In line 421, reference no might be 52.

We apologise for this error; we have corrected the reference in the revised manuscript.

11. The title is ‘autophagy-dependent late ---’. However, late endosomal secretion appears to be parallel to autophagy and autophagy-independent as shown in Fig. 6l.

We initially used the term “autophagy-dependent secretion” in our title because VAMP7-dependent secretion is an adaptive process that increases when degradative autophagy is impaired, as observed in ATG5 KO cells or following Bafilomycin A1 treatment (Fig. 1B–E, Fig. 6A–H). To avoid confusion and better reflect the central message of our study, we have now revised the title to **“Role of VAMP7-dependent late endosomal secretion in endoplasmic reticulum and mitochondrial homeostasis.”**

Reviewer #3 (Remarks to the Author):

The manuscript by Vats et al interrogates the inter-organellar trafficking of proteins that may direct tubular ER and mitochondrial content to multivesicular late endosome (aka MVB) potentially leading to their unconventional secretion. In particular, the paper focuses on VAMP7, which is a vesicular SNARE protein implicated in MVB mediated secretory pathway, and has a role in degradative autophagy, lysosomal secretion, and fusion of mitochondrial vesicles (MDVs) with endosome and lysosome. VAMP7 is not essential for survival in mice, though non-lethal anomalies have been observed, and authors have previously studied the role of VAMP7 in secretory processes in neurons and now expanded this to NRK cells and rat glioblastoma cells. Given the role of stress and autophagy in unconventional secretion the authors' main approach was to generate VAMP7-deficient and ATG5-deficient NRK cells (VAMP7-KO and ATG5-KO respectively) and to compare their secretory properties and impact on mitochondria, ER, as well as exchange of organellar markers between these structures and CD63+ MVBs. The study also examines the transcriptomes of these cell lines wild type and KO cells, and extracts predicted pathways affected by VAMP7 and ATG5 deficiency. Overall, this analysis relies mainly on microscopic colocalization studies, proximity ligation and co-immunoprecipitation of some of the implicated molecular players. Among more notable observations is the evidence that the secretome of NRK cells contains elements of the ER, such as RTN3 isoform, and this is depleted in the case of VAMP7-KO cells, but enhanced in the case of ATG5-KO cells with deficient autophagy pathway. Similar pattern was detected also for mitochondrial proteins such as VDAC leading the authors to propose a role for these pathways in trafficking organellar content to the MVB. Indeed, VAMP7 is found to regulate the protein transport between ER and endosomes, where RTN3 interacts with CD63. This transport is upregulated in settings of mitochondrial and ER stress. Notably, while the cellular responses in the context of VAMP7 and ATG5 perturbations seem interrelated, no viable cells were obtained while genetically targeting both proteins at the same time. Both VAMP7 and ATG5 deficiencies impacted mitochondrial respiration. Finally, authors explore the consequences of VAMP7 and ATG5 targeting in the rat glioblastoma cell line RG2. Interestingly, but not unexpectedly, in this case ATG5 deficiency diminishes tumor growth in the brain, while VAMP7-KO tumors were more necrotic, which the authors attribute to the roles of these proteins in what they refer to as "pathological stress".

We thank the referee for the overall positive view on our findings.

Overall, the study targets and interesting nexus between cellular stress,

1. The most obvious problem is that while nominally the study pertains to unconventional secretion, the nature of this secretion is only scarcely covered in Fig. 1. One may guess that finding ER and mitochondrial proteins in the cellular secretome is a function of release, but this is never formally or directly shown, characterized (MISEV2023) functionally linked to MVB, or explored for function.

We thank the reviewer for this valuable comment. We have now clarified and strengthened our analysis of VAMP7-dependent exosome/small EV release in NRK cells. Specifically, we isolated and characterized EVs from WT, VAMP7KO, and ATG5KO cells (Fig. 1F, G; Supp. Fig. 1B). Using Dynamic Light Scattering-based Nanoparticle Tracking Analysis (NTA), we quantified vesicle size and concentration, showing that VAMP7KO cells are markedly impaired in exosome/small EV secretion compared to WT and ATG5KO cells but not the release of large EVs.

Furthermore, the presence of mitochondrial and ER membrane proteins in the VAMP7-dependent secretome likely reflects the release of vesicular or particulate structures containing these proteins, rather than free proteins per se.

Overall, there is now overwhelming evidence that VAMP7 is required for the biogenesis and secretion of small extracellular vesicles (sEVs) of late-endosomal origin. Early work from our laboratory showed that expressing the isolated longin domain of VAMP7 impairs exocytosis of small EVs²⁵. More recently, we demonstrated that VAMP7 knockout markedly reduces CD63-positive EV release, directly linking VAMP7 to late endosomal/ILV-derived sEV secretion²⁶. Independent confirmation comes from other groups, for example showing that VAMP7 regulates MVB–plasma membrane fusion and exosome release²⁷. Together, these findings fulfill key MISEV2023 criteria²⁸ for assigning EVs to a late-endosomal pathway:

- In all cell types tested, VAMP7 colocalizes to a large extent with CD63^{29,30}
- Use of multiple EV markers, including CD63, the prototypic one,
- Perturbation of a specific molecular component (VAMP7) leading to loss of small EV secretion
- Evidence from independent laboratories
- Morphological characterization of MVBs and EVs.

Importantly, VAMP7 collaborates with SNAP23 in membrane fusion events, and SNAP23 has independently been demonstrated to control small EV/exosome secretion³¹—reinforcing the role of this SNARE pair in late endosomal EV release.

2. Further to this point, the authors place several intriguing hints in the paper, but many are not fully investigated. The case in point is the title of the manuscript: “Role of Autophagy-Dependent Late Endosomal Secretion of ER and Mitochondrial Components in Cell Fitness and Communication”, which suggests studies on cellular fitness and communication neither is investigated in depth.

We appreciate this constructive observation. In our study, the term “cellular fitness” refers to the ability of cells to cope with ER and mitochondrial stress—processes that ultimately reflect organelle and cellular homeostasis. Our results demonstrate that VAMP7-dependent secretion plays a key role in maintaining ER and mitochondrial integrity, as shown by the elevated ER and mitochondrial stress in VAMP7KO cells (Fig. 2C–L).

To make this focus clearer, we have revised the title to:

“Role of VAMP7-dependent late endosomal secretion in endoplasmic reticulum and mitochondrial homeostasis.”

3. The content of ER (Bare et al 2025) and mitochondrial (Ancuzo et al 2021) proteins in extracellular vesicles (EVs) has already been proposed, and to some extent investigated, as have various points of contact and communication between organelles, including EV-generating, VAPA-dependent membrane contact sites (Barman et al 2022). While the current study adds to this body of work, this is mainly in a form of supporting detailed evidence.

We respectfully disagree with the reviewer on this point. While previous studies have described ER and mitochondrial components in EVs, our work provides the first in-depth mechanistic analysis of how VAMP7 regulates their unconventional secretion, and how impairment of this pathway impacts ER and mitochondrial homeostasis.

Our findings do not contradict the existence of membrane contact sites; instead, we propose a complementary mechanism involving ER- and mitochondria-derived vesicles (MDVs). These MDVs are key elements of mitochondrial quality control, and our study reveals that they are secreted in response to autophagy blockade. We have expanded the Discussion to better integrate this perspective and relevant literature.

4. The ‘bifocal’ nature of the present study tackling both VAMP7 and ATG5 is not ideal and makes the work look unfocused. Although these pathways are interlinked and relevant as documented by the absence of viable double knock-out cells, the nature and reason for this

observation is not fully explained. These are also different proteins and pathways that are being compared, especially since their presented connections are often inferential. This, perhaps, overly broad scope results in somewhat descriptive nature of the analysis and the abundance of hypotheses and speculations.

We understand the reviewer's concern. The central focus of our study is the role of VAMP7 in regulating late endosomal secretion of ER and mitochondrial proteins. ATG5KO cells serve as a critical comparator, since they exhibit increased VAMP7-dependent secretion (Fig. 1B–E, Fig. 6A–H). This demonstrates that VAMP7-mediated secretion is adaptive—it becomes upregulated when degradative autophagy is compromised, either genetically (ATG5KO) or pharmacologically (Bafilomycin A1 treatment; Fig. 1B–E, Fig. 6A–H). In parallel, our data show that VAMP7 loss leads to enhanced ER and mitochondrial stress (Fig. 2C–L), directly linking late endosomal secretion to organelle homeostasis.

5. A related point is that while the authors invoke various cellular and pathological “stresses” the study does not have a defined biological direction where these stresses would be relevant for larger functions. For example, the impact of changes in question on specific cellular or tissue phenotypes, such as survival, metabolism, function, are not really explored to any greater depth. The present study, as presented, would unlikely explain the pathological alterations in VAMP7-deficient mice.

We acknowledge this point and agree that fully characterizing all downstream metabolic and functional changes induced by VAMP7 loss lies beyond the scope of the present study. However, our prior publications demonstrate that VAMP7 deficiency alters the cellular sphingolipidome—leading to increased ceramides and GM3s, elevated glycerol, and altered phosphatidylethanolamines^{11,32}—which are mechanistically linked to ER and mitochondrial dysfunction, and potentially to impaired exosomal secretion. Our current findings reveal a novel paracrine mechanism by which cells communicate ER and mitochondrial stress, relevant to the cancer context. We have incorporated this information into the Discussion to strengthen the conceptual framework.

6. This point is also reflected in the study on glioblastoma, which is woefully inadequate and of uncertain validity. While the observations of altered tumor growth, necrosis and progression are intriguing they are not mechanistically explained. There is no direct evidence as to the role of cell-autonomous or secretory events in these changes, and neither is there any convincing documentation that VAMP7 is a major force in the context of highly heterogeneous landscape of human glioblastoma (Wen et al 2020), let alone “tumor microenvironment”. Although the authors include Kaplan Meier curves documenting some difference in survival of glioblastoma patients with VAMP7-high and VAMP7– low tumor characteristics, this may represent the effect of broader molecular subgroups rather than VAMP7 itself. Why would patients with low VAMP7 expression have a survival advantage? Linking this to experiments with a single rodent glioma cell line is probably an overextension.

We appreciate this critic and have significantly expanded our in vivo analyses and discussion. We now include new experiments assessing macrophage and neutrophil infiltration in glioblastoma tumors.

We found that tumors derived from ATG5KO RG2 cells were smaller than those from WT or VAMP7KO cells, consistent with previous reports that autophagy deficiency reduces tumor growth^{21,22}. This likely reflects diminished metabolic adaptability under stress.

VAMP7KO tumors showed a more profound reduction in macrophage recruitment and increased necrosis. This phenotype likely stems from impaired VAMP7-dependent secretion, defective stress adaptation, and reduced clearance of lipidic and protein debris.

We also observed that macrophages were restricted to the periphery of VAMP7KO tumors, while neutrophil infiltration was unaffected—consistent with the distinct biophysical requirements of these immune populations (Fig. 7L–N; Supp. Fig. 7I–J).

Overall, our data suggest that late endosomal secretion regulates macrophage recruitment and immune communication within tumors. Reduced ER- and mitochondria-derived protein secretion in VAMP7KO cells likely limits the release of stress-related cues that attract macrophages to hypoxic or necrotic regions. Together with clinical correlations showing that low VAMP7 or RTN3 expression associates with poorer prognosis (Fig. 7A–C; Supp. Fig. 7A–H), we propose that VAMP7-dependent secretion is a key regulator linking cellular stress signaling to immune modulation in the tumor microenvironment. We are now discussing this point in detail.

7. The authors make frequent and consequential references to their previous work which the reader is not obligated to follow. The paper is also written in a somewhat cryptic and meandering manner, such that it is often difficult to extract the line of authors' reasoning, a sense that is amplified by suboptimal expressions and sometimes overly bold statements.

We appreciate that the readers do not have to check prior references, including those from our prior work. However, this study is grounded in a large body of work which must be cited as a matter of scientific rigor.

For example:

- Page 5: Experiments with NRK cells are hardly a basis “generalization” of VAMP7 effects
- Page 5: Expression: “localization of CD63 to the perinuclear ration found in WT cells” – should this be “perinuclear compartment”?
- Page 7: “These results showed that VAMP7KO does induce major autophagy alteration” – should this be: “does not”?
- Page 10: Exosomal CD63 often appears as smear in many studies, so the speculation as to its role is unfounded.
- Page 12: The statement: “We found a tendency towards larger tumors for VAMP7KO RG2 cells which was not visible in fixed tumors, likely due to removal of tumor interstitial fluid by fixation (Fig 7D, E)” exemplifies ambivalent and speculative comments that add little to the content.

We thank the reviewer for these editorial remarks. We have carefully revised the text to improve clarity, conciseness, and flow, ensuring that our reasoning is easier to follow and that prior references are used only where necessary to support key points. We also corrected typographical errors and imprecise expressions noted by the reviewer, including:

- Page 5: removed “generalization” and corrected “perinuclear ratio” to “perinuclear compartment”;
- Page 7: corrected to “VAMP7KO does **not** induce major autophagy alterations”;
- Page 10: rephrased discussion of exosomal CD63 to remove speculation; we have clarified the issue related to its glycosylation;

- Page 12: revised the speculative comment about tumor size and fixation to improve clarity.

Reviewer #4 (Remarks to the Author):

We thank the reviewer for their efforts in co-reviewing our manuscript and for providing critical feedback.

1. Fasshauer, D., Sutton, R. B., Brunger, A. T. & Jahn, R. Conserved structural features of the synaptic fusion complex: SNARE proteins reclassified as Q- and R-SNAREs. *Proc Natl Acad Sci USA* **95**, 15781–15786 (1998).
2. McLelland, G.-L., Lee, S. A., McBride, H. M. & Fon, E. A. Syntaxin-17 delivers PINK1/parkin-dependent mitochondrial vesicles to the endolysosomal system. *J. Cell Biol.* **214**, 275–291 (2016).
3. Grumati, P. *et al.* Full length RTN3 regulates turnover of tubular endoplasmic reticulum via selective autophagy. *eLife* **6**, (2017).
4. Kimura, T. *et al.* Dedicated SNAREs and specialized TRIM cargo receptors mediate secretory autophagy. *EMBO J.* **36**, 42–60 (2017).
5. Yambire, K. F. *et al.* Impaired lysosomal acidification triggers iron deficiency and inflammation in vivo. *eLife* **8**, (2019).
6. Domingues, N. *et al.* Connexin43 promotes exocytosis of damaged lysosomes through actin remodelling. *EMBO J.* **43**, 3627–3649 (2024).
7. Ebner, M. *et al.* Nutrient-regulated control of lysosome function by signaling lipid conversion. *Cell* **186**, 5328-5346.e26 (2023).
8. Jerabkova-Roda, K. *et al.* Peripheral positioning of lysosomes supports melanoma aggressiveness. *Nat. Commun.* **16**, 3375 (2025).
9. Johnson, D. E., Ostrowski, P., Jaumouillé, V. & Grinstein, S. The position of lysosomes within the cell determines their luminal pH. *J. Cell Biol.* **212**, 677–692 (2016).

10. Huang, R. *et al.* Deacetylation of nuclear LC3 drives autophagy initiation under starvation. *Mol. Cell* **57**, 456–466 (2015).
11. Wojnacki, J. *et al.* Role of VAMP7-Dependent Secretion of Reticulon 3 in Neurite Growth. *Cell Rep.* **33**, 108536 (2020).
12. Pols, M. S. & Klumperman, J. Trafficking and function of the tetraspanin CD63. *Exp. Cell Res.* **315**, 1584–1592 (2009).
13. Justo, B. L. & Jasiulionis, M. G. Characteristics of TIMP1, CD63, and β 1-Integrin and the Functional Impact of Their Interaction in Cancer. *Int. J. Mol. Sci.* **22**, (2021).
14. D'Eletto, M., Oliverio, S. & Di Sano, F. Reticulon Homology Domain-Containing Proteins and ER-Phagy. *Front. Cell Dev. Biol.* **8**, 90 (2020).
15. Kitani, S., Berenstein, E., Mergenhagen, S., Tempst, P. & Siraganian, R. P. A cell surface glycoprotein of rat basophilic leukemia cells close to the high affinity IgE receptor (Fc epsilon RI). Similarity to human melanoma differentiation antigen ME491. *J. Biol. Chem.* **266**, 1903–1909 (1991).
16. Cao, L. *et al.* Global site-specific analysis of glycoprotein N-glycan processing. *Nat. Protoc.* **13**, 1196–1212 (2018).
17. Freeze, H. H. & Kranz, C. Endoglycosidase and glycoamidase release of N-linked glycans. *Curr. Protoc. Mol. Biol.* **Chapter 17**, Unit 17.13A (2010).
18. Ageberg, M. & Lindmark, A. Characterisation of the biosynthesis and processing of the neutrophil granule membrane protein CD63 in myeloid cells. *Clin. Lab. Haematol.* **25**, 297–306 (2003).
19. Lippincott-Schwartz, J. *et al.* Brefeldin A's effects on endosomes, lysosomes, and the TGN suggest a general mechanism for regulating organelle structure and membrane traffic. *Cell* **67**, 601–616 (1991).

20. Xu, F. *et al.* COPII mitigates ER stress by promoting formation of ER whorls. *Cell Res.* **31**, 141–156 (2021).
21. Marsh, T. *et al.* Autophagic Degradation of NBR1 Restricts Metastatic Outgrowth during Mammary Tumor Progression. *Dev. Cell* **52**, 591-604.e6 (2020).
22. Wang, J.-B. *et al.* Targeting mitochondrial glutaminase activity inhibits oncogenic transformation. *Cancer Cell* **18**, 207–219 (2010).
23. Liu, H.-S. *et al.* The role of Atg5 gene in tumorigenesis under autophagy deficiency conditions. *Kaohsiung J. Med. Sci.* **40**, 631–641 (2024).
24. Wculek, S. K., Dunphy, G., Heras-Murillo, I., Mastrangelo, A. & Sancho, D. Metabolism of tissue macrophages in homeostasis and pathology. *Cell. Mol. Immunol.* **19**, 384–408 (2022).
25. Proux-Gillardeaux, V., Raposo, G., Irinopoulou, T. & Galli, T. Expression of the Longin domain of TI-VAMP impairs lysosomal secretion and epithelial cell migration. *Biol. Cell* **99**, 261–271 (2007).
26. Filippini, F. *et al.* Secretion of VGF relies on the interplay between LRRK2 and post-Golgi v-SNAREs. *Cell Rep.* **42**, 112221 (2023).
27. Liu, C. *et al.* Identification of the SNARE complex that mediates the fusion of multivesicular bodies with the plasma membrane in exosome secretion. *J. Extracell. Vesicles* **12**, e12356 (2023).
28. Welsh, J. A. *et al.* Minimal information for studies of extracellular vesicles (MISEV2023): From basic to advanced approaches. *J. Extracell. Vesicles* **13**, e12404 (2024).
29. Wang, G. *et al.* Biomechanical Control of Lysosomal Secretion Via the VAMP7 Hub: A Tug-of-War between VARP and LRRK1. *iScience* **4**, 127–143 (2018).

30. Chiaruttini, G. *et al.* The SNARE VAMP7 Regulates Exocytic Trafficking of Interleukin-12 in Dendritic Cells. *Cell Rep.* **14**, 2624–2636 (2016).
31. Verweij, F. J. *et al.* Quantifying exosome secretion from single cells reveals a modulatory role for GPCR signaling. *J. Cell Biol.* **217**, 1129–1142 (2018).
32. Molino, D. *et al.* Role of tetanus neurotoxin insensitive vesicle-associated membrane protein in membrane domains transport and homeostasis. *Cell. Logist.* **5**, e1025182 (2015).

Reviewer #1 (Remarks to the Author):

The authors have adequately addressed most of my initial concerns.

One remaining point I would like to note relates to the discussion of RTN3A and RTN3L. In lines 501–510, the revised text suggests that RTN3A is primarily associated with protein secretion, whereas RTN3L is involved in protein degradation. However, a recent study suggests that RTN3L may also play a role in protein secretion (PMID: 39919755). Therefore, it is recommended that this discussion be either removed or rewritten to better reflect the current understanding.

We thank the reviewer for their efforts to improve our manuscript. We already cited this exciting recent work, “Blocking conventional secretion was previously shown to increase unconventional secretion, at least in the case of CFTR ¹ in the discussion. Following this reviewer’s suggestion, we have now changed to **“Blocking conventional secretion was previously shown to increase unconventional secretion, at least in the case of CFTR, following a mechanism involving RTN3L and ER stress ¹. This might suggest cross-talks between conventional and unconventional secretions yet to be further explored.”**

Reviewer #2 (Remarks to the Author):

This revised manuscript entitled “Role of VAMP7-dependent late endosomal secretion in the control of endoplasmic reticulum and mitochondrial homeostasis” by Vats S et al. has been improved by incorporation of the reviewer’s inquiries. I still have a couple of inquiries.

We thank the reviewer for their critical comments, which helped to improve our manuscript.

1. In the answer to the inquiry 2 of Reviewer 2, the reference number could be 38 instead of 20.

We have checked this point and confirm that the work related to ER stress whorls is 38, not 20, the latter being associated with VAMP7-dependent ATP secretion.

2. In line 441-448, the most important result is the increase of tumor volume of VAMP7 KO cell graft. Then, other findings were described earlier and in more detail.

We agree with this remark and have rewritten the sentence accordingly as follows: **“Direct analysis of fresh and fixed tumors showed that VAMP7KO cells generated considerably larger tumors than ATG5KO cells (Fig. 7F-I). “**

3. In Fig. 7M, it is not clear what the y axis label (%Macrophages in tumor) represents.

%Macrophages in tumor represents the percentage of IBA1-labelled signal intensity measured across the entire tumor volume. **We have introduced this in the legend of Figure 7. We have also added more detail about the quantification.**

QUANTIFICATION

For quantifying tumor area and the ratio of tumor surface to macrophage or neutrophil staining, we used the QPATH software. Tumors were easily delineated in the brains upon HES staining (as illustrated in Figure 3). Based on the delineation of the tumor area, the percentage of staining was quantified in all tumor sections. **This information was added to the manuscript.**

Figure 1 for the reviewers: Illustration of the use of QPATH to determine the areas of relevance for quantification of DAB precipitate signals (catalyzed by HRP coupled secondary antibodies).

4. In Fig. 7N, there was no significant difference of %macrophage in tumor periphery between WT and VAMP7 KO. However, in line 445-6, 'macrophage restriction to the periphery of VAMP7 KO' was mentioned.

Overall, macrophages infiltrated less VAMP7KO than WT tumors. We have rewritten as follows: **“Macrophages invaded much less VAMP7KO tumors than WT tumors but were equally present in the periphery of VAMP7KO compared to WT. ATG5KO tumors were less invasive overall and in the periphery, but this might be related to their much smaller volume. In contrast, neutrophil infiltration remained throughout the tumor region in all cases (Fig. 7L-N, Supp. Fig. 7I-J).”**

5. Even if macrophage was restricted to the periphery, it could be the consequence of tumor necrosis in the central area of the tumor.

We agree with this remark and have added as follows: **“The decreased macrophage infiltration of VAMP7KO tumors could be related to the increased necrosis, since when macrophage recruitment is impaired or delayed, necrotic areas tend to expand and persist”**.

6. Even if tumor size was different between WT & VAMP7, there is no evidence that the difference is due to changes in innate immunity in response to altered release of stress-related molecules.

This reviewer is correct in pointing out this question. Resolving this issue is one of our priorities, but it will require very substantial work, as we would need to test the effect of RG2 cell secretome without grafting the cells themselves. We have rewritten as follows: **“Altogether, our findings support the notion that late endosomal secretion is likely a determinant of macrophage recruitment to tumors. Further experiments should test the effect of RG2 secretome without grafting the cells *in vivo*.”**

7. In lines 552-555, there is no evidence that increased necrotic area is due to reduced macrophage-mediated removal of debris.

We agree with this remark, which goes back to #5. We have rewritten as noted above, being cautious about the conclusion at this point.

8. In Fig. 1H-I, it is not clearly seen whether the distance from the nucleus is increased in VAMP7 KO cells. Probably, broken circular line in Fig. 7H is plasma membrane boundary. It may be helpful to circle nuclei.

The broken lines in Fig. 1H indeed indicate the plasma membrane boundary. We have also edited the micrographs to mark the nucleus.

Reviewer #3 (Remarks to the Author):

The revisions and rebuttals following the prior review of the manuscript by Vats et al address large proportion of initial concerns. It is a nice paper. While one may agree with much of the authors' argumentation as to inter-organellar interactions, and secretory effects of VAMP7 the glioblastoma (GBM) aspect of the study continues to be a stretch for several reasons:

We thank the reviewer for their positive and insightful remarks.

1. A single rodent cell line is a far cry from the enormous heterogeneity of human GBM, for which proper models would include PDX, or human glioma stem cell isolates, and definitely more than one to make any meaningful conclusions. What subtype of brain tumours does the RG2 cell line represent (transcriptome, markers, lineage identity, properties)?

We searched for IDH1 and IDH2 mutations in RG2 cells by RT-PCR and found them to be WT (Table S4). Therefore, the RG2 cell line could be considered a relevant model for mimicking the most aggressive and frequent forms of adult glioblastoma, characterized by high proliferative capacity, extensive necrosis, profound metabolic rewiring, and a highly immunosuppressive microenvironment, compared with IDH-mutant glioma cells². We have added this information when introducing the cell line.

We also tested the expression of VAMP7 in various primary glioblastoma cell lines isolated from dissociated tumors and grown as spheres. These results shown below, demonstrate that VAMP7 is expressed in primary GB lines grown as sphere (these lines were reported previously³) (Figure 2A). In addition, for some lines we also derived the adherent cell counterparts³. We analyzed the expression of VAMP7 using microarray analyses and show that VAMP7 mRNA is globally expressed in primary tumor cells growing as spheres than those growing as adherent cells. These results were comparable to those obtained the initial tumors but globally higher than in the astrocytes or progenitor cells from the mixture (Figure 2B). These results indicate that VAMP7 is expressed in GB tumor cells. These data were not included in this manuscript because they would require referring to cell lines not further used. The data are shown only to the reviewers.

Figure 2 for the reviewers: VAMP7 expression in primary GB lines. A) Expression of VAMP7 mRNA in 12 primary lines grown as spheres as quantified by microarray analyses. B) Expression of VAMP7 mRNA in tumor extracts, primary lines grown as spheres, primary lines grown as adherent cells and non-cancerous cells.

2. The nomenclature of brain tumours used by the authors is extremely confusing. The current (2021) WHO classification moves away from TCGA subtypes, which include considerable proportion of IDH1 mutant tumours, mainly in the proneural subtype. IDH1 mutant tumours are no longer included in the GBM diagnosis and possess vastly different clinical and cellular characteristics than GBM (Wen et al 2020). This is never clarified in the paper, which in Supplemental Figure 7 contains a random mix of low and high grade tumours from different cohorts and of different origins, with names such as “diffuse astro glioma” (?), some of which are simply not in use.

This is true, and the data presented in Figure S7 were not intended to suggest that low expression of VAMP7 was a poor prognostic factor only in glioblastoma, but rather that this observation was also applicable to other primary tumors of the central nervous system. The text has been corrected accordingly on line 450 of the revised manuscript as follows: **“Beyond glioblastoma, low expression of VAMP7 was thus significantly associated with poorer prognosis in a large panel of primary tumors from the central nervous system, likely indicating its involvement in a general mechanism in those tumors.”**

In addition, in Fig. 7 along with the corresponding text and rebuttal the authors single out TCGA Stupp tumours. What are they? Stupp protocol is a standard of care for virtually all newly diagnosed IDH-WT GBMs. Are the data representative of recurrent tumours? Is this the reason why VAMP7 no longer separates survival patterns? Which cells in the tumours express VAMP7?

We agree with this reviewer. The analyses were carried out on a subset of the TCGA database for which all the clinical information and H3K27 methylation data were available⁴. This subset of data comprises 226 patients with homogenous age range, and clinical data. This information was added to this revised manuscript.

The methylation data were also explored to test whether VAMP7 regulation could have occurred through an epigenetic mechanism (which was not the case and was not shown).

This new information was not added to the revised manuscript because they refer to a topic not addressed in detail here. They are shown for the reviewers only.

Can this be inferred from single cell datasets which are readily accessible? Are the authors suggesting that VAMP7 is uniquely linked to brain tumours or is this a pathway of many cancer cells under metabolic stress? The GBM aspect of the present study should be either qualified, as preliminary (which it is) or strengthened and properly presented.

The history behind the association with glioblastoma comes from the fact that the expression of VAMP7 was found to be regulated by ER stress in those tumors. First of all, VAMP7 mRNA can be cleaved in vitro by the ER stress sensor IRE1⁵. When we explored the expression of VAMP7 in GB (GBM-mark cohort as used in Fig 7A in the present manuscript) based on their classification according to IRE1 activity (comprising XBP1 mRNA splicing (X+/-) or mRNA degradation through RIDD (R+/-)) we observed that the expression of VAMP7 mRNA varied in function of IRE1 activity (and likely ER stress elicited in those tumors) and that results were consistent with the observed in vitro cleavage of VAMP7 mRNA by IRE1 (Figure 3).

Figure 3 for the reviewers: Expression of VAMP7 mRNA in tumors from the GBM-Mark cohort and classified according to their endogenous IRE1 activity (XBP1 mRNA splicing (X) or RIDD (R)).

It is likely that the observation made on the role of VAMP7 in GB might be also applicable to other tumor types in which ER and/or mitochondrial homeostasis are challenged, as a contributing factor to a major adaptive pathway.

These data were not included in the manuscript because they pertain to a topic not fully addressed in detail and are shown only to the reviewers.

QUANTIFICATION

For quantifying tumor area and the ratio of tumor surface to macrophage or neutrophil staining, we used the QPATH software. Tumors were easily delineated in the brains upon HES staining (as illustrated in Figure 2). Based on the delineation of the tumor area, the percentage of staining was quantified in all tumor sections. **This information was added to the manuscript.**

Figure 2 for the reviewers: Illustration of the use of QPATH to determine the areas of relevance for quantification of DAB precipitate signals (catalyzed by HRP coupled secondary antibodies).

REFERENCES

1. Song, M. S. *et al.* Tubular ER structures shaped by ER-phagy receptors engage in stress-induced Golgi bypass. *Dev. Cell* (2025) doi:10.1016/j.devcel.2025.01.011.
2. Bayona, C., Randelović, T. & Ochoa, I. Tumor Microenvironment in Glioblastoma: The Central Role of the Hypoxic-Necrotic Core. *Cancer Lett.* 218216 (2025) doi:10.1016/j.canlet.2025.218216.
3. Avril, T. *et al.* CD90 expression controls migration and predicts dasatinib response in glioblastoma. *Clin. Cancer Res.* **23**, 7360–7374 (2017).
4. Xiang, Y., Zhang, C.-Q. & Huang, K. Predicting glioblastoma prognosis networks using weighted gene co-expression network analysis on TCGA data. *BMC Bioinformatics* **13 Suppl 2**, S12 (2012).
5. Lhomond, S. *et al.* Dual IRE1 RNase functions dictate glioblastoma development. *EMBO Mol. Med.* **10**, (2018).